# Shift-and-Sum Quantization for Visual Autoregressive Models

**Jaehyeon Moon**[1,2]      **Bumsub Ham**[1]*
[1]Yonsei University      [2]Articron
{jhmn,bumsub.ham}@yonsei.ac.kr

## Abstract

Post-training quantization (PTQ) enables efficient deployment of deep networks using a small set of data. Its application to visual autoregressive models (VAR), however, remains relatively unexplored. We identify two key challenges for applying PTQ to VAR: (i) large reconstruction errors in attention–value products, especially at coarse scales where high attention scores occur more frequently; and (ii) a discrepancy between the sampling frequencies of codebook entries and their predicted probabilities due to limited calibration data. To address these challenges, we propose a PTQ framework tailored for VAR. First, we introduce a shift-and-sum quantization method that reduces reconstruction errors by aggregating quantized results from symmetrically shifted duplicates of value tokens. Second, we present a resampling strategy for calibration data that aligns sampling frequencies of codebook entries with their predicted probabilities. Experiments on class-conditional image generation, inpainting, outpainting, and class-conditional editing show consistent improvements across VAR architectures, establishing a new state of the art in PTQ for VAR. Project page is available at: http://cvlab.yonsei.ac.kr/projects/Shift-and-Sum/

## 1 Introduction

With the advancement of large language models (LLM), researchers have increasingly adopted autoregressive (AR) modeling—i.e., predicting the next token in a sequence—for a variety of generative tasks. Recently, visual autoregressive models (VAR) (Tian et al., 2024) have emerged as a powerful alternative to state-of-the-art generative models (e.g., diffusion models (Ho et al., 2020), GANs (Goodfellow et al., 2014)), by replacing conventional next-token generation with a next-scale generation strategy, making them increasingly popular in the field of computer vision. However, the use of multiple transformer blocks and their iterative generation process across multiple scales require significant computational resources (e.g., memory, FLOPs). As a result, network compression techniques (e.g., pruning, quantization) have gained significant attention to enable their deployment on practical devices. Network quantization, which converts the weights and activations of a model into low-bit representations for efficient inference, is typically classified into two categories: quantization-aware training (QAT) and post-training quantization (PTQ). QAT methods (Rastegari et al., 2016; Esser et al., 2020) apply quantizers to weights and activations and retrain the model using full training dataset, which is computationally intensive. In contrast, PTQ methods (Banner et al., 2019; Shang et al., 2023) calibrate quantization parameters (e.g., scale, zero-point) using a small subset of training data only, allowing for rapid and resource-efficient deployment.

VAR consists of a VAR transformer and a multi-scale vector quantized variational autoencoder (VQVAE) (Van Den Oord et al., 2017), referred to as the VAR tokenizer. It employs a hierarchical generation process that begins with a start token at the coarsest scale. At each scale, the VAR transformer takes token maps from previous scales as input and predicts a probability distribution over entries of the VQVAE codebook for each position. At each position, a single entry is sampled from the predicted probabilities to construct the token map at current scale. This process iterates across scales, progressively refining the representation to higher resolutions. After completing all scales, the final token map is passed to the VQVAE decoder to generate the image.

---

*Corresponding author.

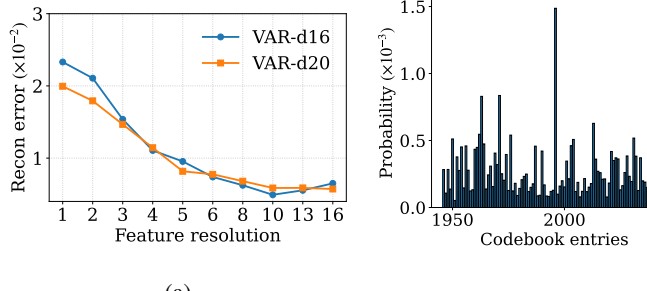 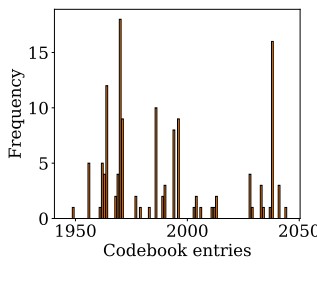

(a)                                                    (b)

Figure 1: (a) Visualization of reconstruction errors for the multiplication of attention scores and value tokens across scales, for a particular transformer block of VAR-d16 and VAR-d20; (b) Comparison between predicted probabilities and sampling frequencies of codebook entries, aggregated from 256 calibration samples in VAR-d24. Note that we show 100 codebook entries only for visualization. We can see that attention-value products tend to incur large reconstruction errors at coarse scales, and sampling frequencies of codebook entries do not align with their predicted probabilities.

Although PTQ has shown promising results on conventional generative models, e.g., diffusion models (Ho et al., 2020), its application to VAR remains relatively unexplored. We identify two key challenges in quantizing VAR (Fig. 1): (1) Multiplication between attention scores and value tokens in VAR transformer incurs significant errors after quantization, especially at low resolutions. We find that quantization errors are amplified for value tokens with large attention scores, which occur more frequently at low resolutions due to the reduced number of tokens. To this end, we introduce a shift-and-sum quantization technique, that duplicates the value tokens associated with large attention scores, applies symmetric shifts around the original value, and averages the quantized results to suppress quantization errors. (2) In the calibration set, the sampling frequency of each codebook entry—i.e., the number of times it is sampled across all calibration samples—often diverges from the predicted probabilities aggregated over token positions, due to the extremely limited number of available samples in PTQ. This mismatch biases the quantization parameters toward an inaccurate distribution, degrading quantization performance. To this end, we introduce a calibration data resampling technique that aligns the frequencies with predicted probabilities in the calibration data by reallocating tokens from oversampled to undersampled codebook entries.

Our contributions are summarized as follows:

- We identify two key challenges specific to quantizing VAR: (1) significant quantization errors arising from the multiplication between attention scores and value tokens, especially at coarse scales; and (2) a mismatch between the predicted probabilities over the entries of VQVAE codebook and their sampling frequencies during calibration.

- We propose a shift-and-sum quantization technique that reduces quantization error in attention-value products by aggregating quantized results from symmetrically shifted value tokens.

- We introduce a calibration data resampling technique to align the sampling frequencies of codebook entries with their predicted probabilities in the calibration data.

- We establish a new state of the art on class-conditional image generation across VAR architectures, with consistent improvements on the tasks of image inpainting, outpainting, and class-conditional editing.

## 2   RELATED WORKS

### 2.1   VAR

AR models (Van den Oord et al., 2016; Ramesh et al., 2021; Sun et al., 2024) generate images by predicting each token once at a time in a raster-scan order, conditioning each step on previously generated tokens. The generation process captures fine-grained dependencies between tokens but incurs substantial overhead especially for high-resolution images (Razavi et al., 2019). In contrast, VAR (Tian et al., 2024) adopts a coarse-to-fine generation strategy: it generates token maps progressively from coarse to fine scales, with each scale conditioned on the token maps of preceding scales. This hierarchical design enhances structural coherence and improves sampling efficiency compared

to conventional AR models. ControlVAR (Li et al., 2024) extends this framework to conditional image generation, where the model takes a control map (e.g., segmentation mask) as input and generates an image conditioned on them. Infinity (Han et al., 2025) introduces a bitwise tokenizer that encodes each token as a binary sequence, allowing the number of possible representations to grow exponentially with the sequence length. CoDe (Chen et al., 2025) generates token maps using a large drafter model at coarse scales and a small refiner model at fine scales. Despite their strong generative capabilities, these models remain computationally demanding due to their hierarchial generation process and multiple transformer blocks.

## 2.2 Network quantization

Network quantization reduces the bit-widths of weights and activations in a model to enable efficient inference. QAT methods incorporate quantizers into the weights and activations of the model and retrain it using differentiable approximations of the rounding function (e.g., Esser et al. (2020)). Despite their strong performances, QAT requires access to full training dataset and incurs substantial computational cost. In contrast, PTQ adjusts quantization parameters using only a subset of training data, enabling fast and data-efficient deployment. Early works calibrate quantization intervals by minimizing quantization error (Banner et al., 2019) or task-specific losses (Nahshan et al., 2021). Another line of research (Nagel et al., 2020; Li et al., 2021; Lee et al., 2023) focuses on optimizing the rounding function for network weights instead of using standard rounding-to-nearest schemes.

To date, a number of PTQ techniques have been developed for transformer architectures. For example, PTQ4ViT (Yuan et al., 2022) proposes twin uniform quantizers to address the long-tailed distributions of softmax attentions and activations after GELU (Hendrycks & Gimpel, 2016) non-linearity. RepQ-ViT (Li et al., 2023b) proposes to consider inter-channel scale variations of activations after LayerNorm, and exploits a scale reparameterization technique that adjusts the affine factors of LayerNorm and the weights of subsequent fully-connected (FC) layers. IGQ-ViT (Moon et al., 2024) observes that input activations of FC layers vary drastically across input instances, and splits the channels of activation maps dynamically into multiple groups, quantizing the activation values within each group with a shared quantizer. ERQ (Zhong et al., 2025) proposes to adjust the rounding function for weights to minimize the quantization errors induced from activation quantization.

Recent methods have extended PTQ to diffusion models (Ho et al., 2020), which synthesize images by iteratively applying a denoising process over time steps. For example, PTQ4DM (Shang et al., 2023) proposes to sample more images from later time steps for calibration, which are typically closer to real images. Q-diffusion (Li et al., 2023a) observes that concatenation of activations from different layers can result in significant inter-channel scale variation, and proposes to quantize activations before concatenation to mitigate this issue. TFMQ-DM (Huang et al., 2024) shows that quantizing temporal embedding layers can distort temporal information in diffusion models, and proposes to calibrate them separately from other layers. AccuQuant (Lee et al., 2025) groups consecutive time steps and minimizes the discrepancy between the quantized and full-precision outputs within each group, thereby reducing the accumulation of quantization error across time steps.

Despite these advances, applying PTQ to VAR remains largely unexplored. To the best of our knowledge, LiteVAR (Xie et al., 2024) is the only PTQ method for VAR. It demonstrates that quantizing FC layers following the GELU non-linearity leads to substantial performance degradation, and retains them in full-precision. However, this approach introduces considerable computational overhead and requires hardware support for mixed-precision arithmetic. In contrast, our approach quantizes all tensors involved in matrix multiplications—including the weights and activations for every FC layer, as well as the activations in self-attention mechanisms (i.e., query, key, value, and softmax attention)—using the same bit-width, allowing for efficient deployment on conventional hardware.

## 3 Method

### 3.1 Preliminaries

**Uniform quantizer.** Given a floating-point value $x$ and quantization bit-width $b$, the uniform quantizer maps $x$ to a $b$-bit integer as follows:

$$\hat{x} = \text{clip}(\lfloor \frac{x}{s} \rceil + z, \ 0, \ 2^b - 1), \tag{1}$$

where the step size $s$ and zero-point $z$ are defined as:

$$s = \frac{\max(x) - \min(x)}{2^b - 1}, \quad z = \text{clip}(\lfloor -\frac{\min(x)}{s} \rceil, 0, 2^b - 1). \tag{2}$$

We denote $\lfloor . \rceil$ as a rounding function, and $\text{clip}(., m, n)$ as a clipping function with lower and upper bounds of $m$ and $n$, respectively. The quantized output is then computed as:

$$Q(x; s, z) = s(\hat{x} - z). \tag{3}$$

**Log2 quantizer.** The log2 quantizer discretizes $x$ with a bit-width of $b$ as follows:

$$\hat{x} = \text{clip}(\lfloor -\log_2 \frac{x}{s} \rceil, 0, 2^b - 1), \tag{4}$$

where the scale $s$ is typically set to $\max(x)$. The quantized output is obtained as:

$$Q(x; s) = s 2^{-\hat{x}} \tag{5}$$

where both the logarithm and exponential base-2 operations can be efficiently implemented using bit-shift operations. Following (Lin et al., 2022; Li et al., 2023b), we exploit log2 quantizers for softmax attentions, and uniform quantizers for all other weights and activations.

## 3.2 PROPOSED METHOD

In this section, we introduce our PTQ framework for VAR, consisting of two components: (1) *Shift-and-sum quantization* (Sec. 3.2.1), which reduces reconstruction errors in attention–value products of VAR transformer; and (2) *calibration data resampling* (Sec. 3.2.2), which alleviates the mismatch between sampling frequencies of codebook entries and their predicted probabilities during calibration.

### 3.2.1 SHIFT-AND-SUM QUANTIZATION

**Reconstruction error across scales.** VAR exploits an autoregressive, multi-scale generation process that begins with coarse token maps and progressively generates ones at finer scales. We observe that multiplication between attention scores and value tokens exhibits particularly large reconstruction errors at coarse scales (Fig. 1(a)). Errors introduced at these early stages are especially detrimental, since they propagate through subsequent scales, become amplified in the final outputs, and ultimately lead to significant performance degradation after quantization.

We hypothesize that the large quantization error at coarse scales arises from the presence of *attentive tokens*, which we define as value tokens associated with high attention scores. To formalize this, we consider the multiplication between attention scores $\mathbf{a} \in \mathbb{R}^T$ and value tokens $\mathbf{V} \in \mathbb{R}^{T \times d}$, where we denote by $T$ and $d$ the number of value tokens and hidden dimensions, respectively. Both are subject to quantization noise, with $\boldsymbol{\epsilon}^a \in \mathbb{R}^T$ for the attention scores and $\boldsymbol{\epsilon}^v \in \mathbb{R}^{T \times d}$ for the value tokens. The quantized output can then be represented as:

$$\sum_{i=1}^T (a_i + \epsilon_i^a)(\mathbf{v}_i + \boldsymbol{\epsilon}_i^v), \tag{6}$$

where we denote $a_i \in \mathbb{R}$ and $\mathbf{v}_i = \mathbf{V}_{i,:} \in \mathbb{R}^d$ as the $i$-th attention score and value token, respectively, and $\epsilon_i^a \in \mathbb{R}$ and $\boldsymbol{\epsilon}_i^v \in \mathbb{R}^d$ are their corresponding quantization noises. Note that we use log2 quantizers for attention scores. Since they discretize values on a logarithmic scale, the quantization noise $\epsilon_i^a$ increases w.r.t. the magnitude of $a_i$ (Kofman, 2009). The reconstruction error can then be expressed as:

$$\delta = \sum_{i=1}^T (a_i + a_i \tilde{\epsilon}_i^a)(\mathbf{v}_i + \boldsymbol{\epsilon}_i^v) - a_i \mathbf{v}_i = \sum_{i=1}^T a_i (\tilde{\epsilon}_i^a \mathbf{v}_i + \tilde{\epsilon}_i^a \boldsymbol{\epsilon}_i^v + \boldsymbol{\epsilon}_i^v), \tag{7}$$

where we define $\tilde{\epsilon}_i^a = \epsilon_i^a / a_i$. Assuming that $\{\tilde{\epsilon}_i^a\}_{i=1}^T$ and $\{\boldsymbol{\epsilon}_i^v\}_{i=1}^T$ are independent, zero-mean random variables with variances $\sigma_a^2$ and $\sigma_v^2$, respectively, the variance of the reconstruction error $\delta$ can be obtained as:

$$\text{Var}[\delta] = \sum_{i=1}^T a_i^2 \big( \sigma_a^2 \|\mathbf{v}_i\|_2^2 + d(\sigma_a^2 \sigma_v^2 + \sigma_v^2) \big). \tag{8}$$

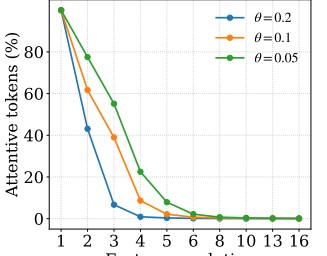 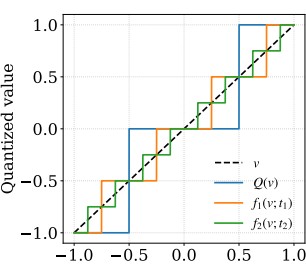 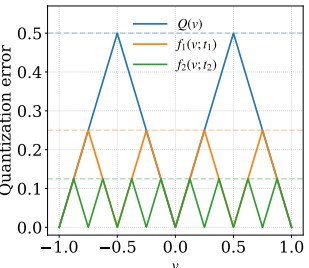

Figure 2: Percentage of tokens with average attention scores exceeding a threshold $\theta$ in VAR-d16 (left); Visualization of quantization kernel $f_n(v; t_n)$ across different kernel orders $n$ (middle); Quantization error of the kernel (i.e., $|v - f_n(v; t_n)|$) across different kernel orders $n$ (right). We present the results for multiple values of $\theta$, and we set $s = 1$, $t_1 = 1/4$, and $t_2 = 1/8$. We visualize the upper bounds of quantization errors with horizontal stripes of corresponding colors.

A detailed derivation of Eq. 8 is provided in Appendix B. We observe that the variance of the reconstruction error increases in the presence of high attention scores, given that their sum is constrained to 1. We observe in Fig. 2(left) that high attention scores are more common at coarse scales, where they are distributed on fewer tokens, leading to large quantization errors.

**Shift-and-sum quantization for attentive tokens.** We propose a shift-and-sum quantization framework to mitigate reconstruction errors in the multiplication between attention scores and value tokens. To this end, we introduce a *quantization kernel* that effectively reduces quantization errors at the cost of extra operations (e.g., bit-shift, summation). We then apply the kernel only to attentive tokens, where the kernel orders are assigned adaptively according to their attention scores. *Quantization kernel.* For a scalar value $v$, we define the kernel as follows (Fig. 2(middle)):

$$f_n(v; t_n) = \frac{1}{2n} \sum_{k=-n}^{n-1} Q\big(v + (2k+1)t_n\big), \tag{9}$$

where we define $n$ and $t_n$ as the order and shift factor for the kernel. The upper bound of the quantization error for $f_n(v; t_n)$ is inversely proportional to $n$, as shown below.

**Theorem 1.** *The quantization error of $f_n(v; t_n)$ defined in Eq. 9 satisfies*

$$|v - f_n(v; t_n)| \le \frac{s}{4n}, \tag{10}$$

*for $t_n = s/4n$, where $s$ is the step size of $Q(\cdot)$. Furthermore, this bound is tight: for arbitrary choices of shift factor $t_n$, the error bound cannot be smaller than $s/4n$.*

We provide the proofs in Appendix C, and visualize the quantization error of $f_n(v; t_n)$ in Fig. 2(right). Following Theorem 1, we set $t_n$ to $s/4n$ for a kernel of order $n$. For a vector $\mathbf{v}$, the kernel is applied independently to each component.

*Application to attention-value products.* Since the quantization kernel in Eq. 9 introduces additional computations, we apply the kernel only to attentive tokens, which contribute most to the reconstruction error. In practice, the kernel can be efficiently implemented by duplicating attentive tokens, shifting their values symmetrically, and aggregating the quantized results. Specifically, we compute the average attention score for each value token as follows:

$$\text{Score}(\mathbf{v}_i) = \frac{1}{T'} \|\boldsymbol{\alpha}_i\|_1, \tag{11}$$

where $T'$ is the number of query tokens, and $\boldsymbol{\alpha}_i = \mathbf{A}_{:,i}$ is the $i$-th column of the attention matrix $\mathbf{A} \in \mathbb{R}^{T' \times T}$. Based on these scores, we identify attentive tokens and quantize them with the kernel, while applying standard quantizers to the remaining tokens:

$$\mathbf{A}\mathbf{V} = \sum_{i=1}^{T} \boldsymbol{\alpha}_i \mathbf{v}_i^\top \approx \sum_{i \in H} Q_a(\boldsymbol{\alpha}_i) f_n(\mathbf{v}_i; \frac{s_v}{4n})^\top + \sum_{i \notin H} Q_a(\boldsymbol{\alpha}_i) Q_v(\mathbf{v}_i)^\top, \tag{12}$$

where $Q_a(\cdot)$ and $Q_v(\cdot)$ are the quantizers for attention scores and value tokens, and $H$ and $s_v$ are the set of attentive tokens and the step size for $Q_v(\cdot)$, respectively. The first term can be further

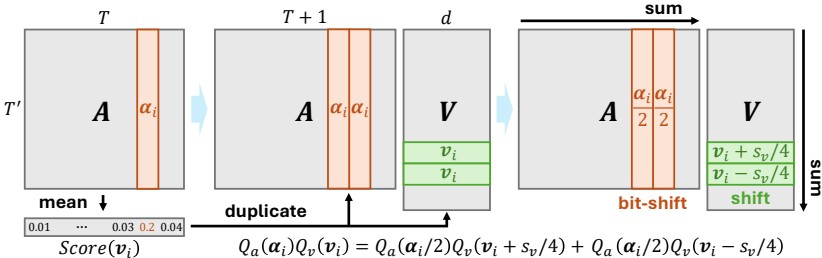

Figure 3: Visualization of attention-value products using shift-and-sum quantization for $n = 1$. We implement the quantization kernel in Eq. 9 by duplicating attentive tokens, and aggregating quantized results from their symmetrically shifted counterparts.

decomposed as:

$$Q_a(\boldsymbol{\alpha}_i) f_n(\mathbf{v}_i; \frac{s_v}{4n})^\top \approx \sum_{k=-n}^{n-1} Q_a\left(\frac{\boldsymbol{\alpha}_i}{2n}\right) Q_v(\mathbf{v}_i + (2k+1)\frac{s_v}{4n})^\top. \tag{13}$$

We illustrate in Fig. 3 the multiplication of attention scores and value tokens for $n = 1$. We restrict the kernel order $n$ to powers of two (e.g., 1,2,4), since dividing attention scores by such values could be implemented as bit-shift operations, enabling efficient computation of $Q_a(\boldsymbol{\alpha}_i/2n)$.

*Adaptive kernel order.* We apply quantization kernels only to tokens whose average attention scores exceed a threshold $\theta$, chosen to satisfy a BOP constraint (see Sec. 4.1). For such tokens, we choose the smallest kernel order that brings the average attention scores below $\theta$. Specifically, for tokens with $\mathrm{Score}(\mathbf{v}_i) > \theta$, we set their orders as $2^{\hat{n}_i - 1}$, where $\hat{n}_i$ is defined as:

$$\hat{n}_i = \left\lceil \log_2 \frac{\mathrm{Score}(\mathbf{v}_i)}{\theta} \right\rceil, \tag{14}$$

where we denote by $\lceil \cdot \rceil$ the ceiling operation. We can see from Eq. 13 that quantizing $\mathbf{v}_i$ with a kernel of order $n$ scales the attention scores $\boldsymbol{\alpha}_i$ by a factor of $1/2n$; with a kernel order of $2^{\hat{n}_i - 1}$ the average attention score is reduced below $\theta$ as follows:

$$\frac{1}{T'} \left\| \frac{\boldsymbol{\alpha}_i}{2^{\hat{n}_i}} \right\|_1 = \frac{\mathrm{Score}(\mathbf{v}_i)}{2^{\hat{n}_i}} \leq \theta, \tag{15}$$

Note that $2^{\hat{n}_i} \geq \mathrm{Score}(\mathbf{v}_i)/\theta$ by the definition of $\hat{n}_i$.

**Comparison to mixed-precision quantization.** An alternative to our method is to allocate higher bit-widths to attentive tokens, which can also reduce reconstruction errors in attention–value products. Consider a matrix multiplication between attention scores $\{\boldsymbol{\alpha}_i\}_{i=1}^T$, quantized with a step size of $s_a$, and value tokens $\{\mathbf{v}_i\}_{i=1}^T$, quantized with step sizes of $\hat{s}_v$ and $s_v$ for attentive tokens and non-attentive ones. The matrix multiplication can be represented as follows (zero-points omitted for clarity):

$$\sum_{i=1}^T \boldsymbol{\alpha}_i \mathbf{v}_i^\top \approx \sum_{i \in H} Q_a(\boldsymbol{\alpha}_i) \hat{Q}_v(\mathbf{v}_i)^\top + \sum_{i \notin H} Q_a(\boldsymbol{\alpha}_i) Q_v(\mathbf{v}_i)^\top$$

$$= \sum_{i \in H} s_a \hat{s}_v \, 2^{-\hat{\boldsymbol{\alpha}}_i} \, \hat{\mathbf{v}}_i^\top + \sum_{i \notin H} s_a s_v \, 2^{-\hat{\boldsymbol{\alpha}}_i} \, \hat{\mathbf{v}}_i^\top, \tag{16}$$

where we denote $\hat{Q}_v(.)$ as a higher-bit quantizer for attentive tokens. $\hat{\boldsymbol{\alpha}}_i$ and $\hat{\mathbf{v}}_i$ are integer values obtained by applying Eq. 4 to $\boldsymbol{\alpha}_i$ and Eq. 1 to $\mathbf{v}_i$, respectively. Computing Eq. 16 requires the summation of two floating-point matrices, which could only be implemented with specialized hardware support. In contrast, our method does not require multiple quantizers for a single matrix, thereby avoiding floating-point summations between different attention-value products.

### 3.2.2 CALIBRATION DATA RESAMPLING

**Mismatch between probabilities and frequencies.** We construct token maps by randomly sampling codebook entries according to their predicted probabilities during calibration. Since PTQ

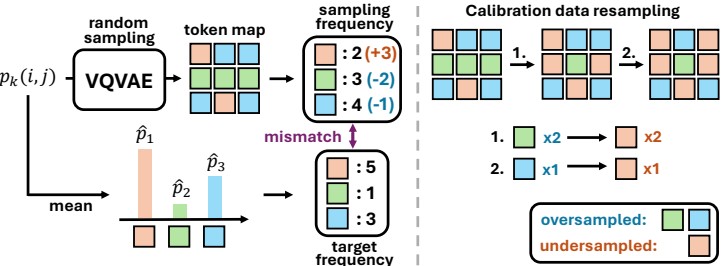

Figure 4: Illustration of the calibration data resampling technique, which reassigns tokens from oversampled codebook entries to undersampled ones, thereby aligning the sampling frequencies with the predicted probabilities aggregated over calibration samples.

typically allows only a small number of calibration samples, this random sampling process incurs a high variance: the sampling frequencies of codebook entries often deviate significantly from their predicted probabilities aggregated over calibration samples (Fig. 1(b)). This problem is exacerbated by the large vocabulary size of VQVAE codebooks (e.g., 4096 entries in VAR-d16), which makes it infeasible that limited calibration data can accurately reflect the underlying probability distribution.

**Probability matching for calibration data generation.** We propose a probability matching strategy to align the sampling frequencies of codebook entries with their predicted probabilities (Fig. 4). Specifically, we first generate calibration samples by randomly sampling token maps according to the predicted probability distributions. We then compute the mean probability of each codebook entry, aggregated over token positions, as follows:

$$\hat{p}_k = \frac{1}{NT} \sum_{i=1}^{N} \sum_{j=1}^{T} p_k(i,j), \tag{17}$$

where we denote by $p_k(i,j)$ the predicted probability of selecting the $k$-th codebook entry for the $j$-th token in the $i$-th calibration sample, and $N$ and $T$ are the number of calibration samples and tokens per sample, respectively. Based on $\hat{p}_k$, we define the target frequency of $k$-th entry as:

$$t_k = NT\hat{p}_k. \tag{18}$$

We define the sets of oversampled and undersampled entries as

$$O = \{k : s_k - t_k \geq 1\}, \quad U = \{k : t_k - s_k \geq 1\}, \tag{19}$$

corresponding to entries sampled more or less frequently than their target frequency. We denote by $s_k$ the sampling frequency of the $k$-th codebook entry. For tokens assigned to entries in $O$, we reassign them to entries in $U$, where the probability for reassignment is defined as follows:

$$\tilde{p}_k(i,j) = \begin{cases} \dfrac{p_k(i,j)}{\sum_{k \in U} p_k(i,j)}, & k \in U, \\ 0, & k \notin U. \end{cases} \tag{20}$$

This process is repeated until $s_k - t_k$ tokens are reassigned for the $k$-th codebook entry, thereby enforcing consistency between its frequency and probability.

# 4 EXPERIMENTS

## 4.1 IMPLEMENTATION DETAILS

We evaluate our method on both VAR (Tian et al., 2024) and its text-to-image extension, Infinity (Han et al., 2025). For conventional VAR models, we consider four tasks: class-conditional image generation, inpainting, outpainting, and class-conditional editing. For class-conditional image generation, we use VARs of varying depths (16, 20, 24, and 30) for evaluating our method. We generate 50 images for each class in ImageNet (Deng et al., 2009), which includes 1.2M images for training, and 50K images for validation. We evaluate the generated images using standard benchmarks (e.g., IS (Salimans et al., 2016), FID (Heusel et al., 2017)). To further validate our method,

Table 1: Quantitative comparison of quantizing VAR with various methods for the task of conditional image generation on ImageNet (Deng et al., 2009). We denote by W/A the bit-widths of weights (W) and activations (A), respectively. We report IS, FID, and FID2FP16 for VARs of varying depths. Note that we have re-implemented LiteVAR (Xie et al., 2024) to ensure consistency with the evaluation settings provided in the work of (Tian et al., 2024).

| #Bits (W/A) | Methods | VAR-d16 | | | VAR-d20 | | | VAR-d24 | | | VAR-d30 | | |
|---|---|---|---|---|---|---|---|---|---|---|---|---|---|
| | | IS | FID | FID2FP16 | IS | FID | FID2FP16 | IS | FID | FID2FP16 | IS | FID | FID2FP16 |
| 16/16 | Full-precision | 274.8 | 3.41 | - | 302.9 | 2.73 | - | 313.2 | 2.17 | - | 303.5 | 1.97 | - |
| 6/6 | BRECQ (Li et al., 2021) | 202.9 | 5.56 | 4.29 | 239.5 | 3.23 | 2.72 | 261.4 | 2.73 | 2.44 | 250.9 | 3.38 | 2.55 |
| | LiteVAR (Xie et al., 2024) | 212.1 | 5.17 | 4.05 | 247.8 | 3.13 | 2.50 | 261.8 | 2.76 | 2.43 | 258.5 | 3.27 | 2.48 |
| | Ours | 213.5 | 4.46 | 3.14 | 249.5 | 3.01 | 2.34 | 264.8 | 2.59 | 2.30 | 262.7 | 2.88 | 2.09 |
| | Ours+LiteVAR | **226.1** | **4.08** | **2.64** | **253.3** | **2.89** | **2.09** | **269.9** | **2.38** | **1.97** | **266.8** | **2.66** | **1.85** |
| 4/8 | BRECQ (Li et al., 2021) | 203.8 | 6.07 | 4.76 | 226.2 | 4.21 | 3.82 | 261.7 | 2.90 | 2.23 | 270.4 | 2.91 | 1.96 |
| | LiteVAR (Xie et al., 2024) | 217.8 | 5.56 | 3.85 | 242.6 | 3.58 | 2.93 | 270.4 | 2.72 | 2.19 | 273.4 | 2.85 | 2.03 |
| | Ours | 212.5 | 5.71 | 4.05 | 241.0 | 3.71 | 2.98 | 275.8 | 2.60 | 1.91 | 276.8 | 2.71 | 1.81 |
| | Ours+LiteVAR | **227.8** | **4.83** | **3.00** | **256.4** | **3.32** | **2.32** | **285.6** | **2.50** | **1.73** | **279.4** | **2.60** | **1.76** |
| 4/6 | BRECQ (Li et al., 2021) | 145.6 | 11.16 | 10.57 | 189.1 | 6.74 | 6.62 | 215.8 | 4.89 | 4.82 | 221.4 | 5.09 | 4.17 |
| | LiteVAR (Xie et al., 2024) | 158.1 | 9.93 | 9.09 | 205.6 | 5.23 | 5.06 | 225.4 | 4.23 | 4.04 | 218.9 | 5.41 | 4.60 |
| | Ours | 162.1 | 9.20 | 8.44 | 202.2 | 5.36 | 5.09 | 230.7 | 3.96 | 3.90 | 227.8 | 4.59 | 3.74 |
| | Ours+LiteVAR | **189.3** | **6.62** | **5.28** | **219.2** | **4.46** | **3.98** | **248.1** | **3.32** | **3.01** | **235.3** | **4.43** | **3.56** |
| 4/4 | BRECQ (Li et al., 2021) | 67.6 | 33.03 | 32.82 | 94.1 | 23.03 | 22.42 | 119.0 | 16.89 | 17.07 | 125.3 | 15.71 | 14.66 |
| | LiteVAR (Xie et al., 2024) | 66.9 | 35.87 | 36.67 | 100.7 | 21.71 | 21.10 | 128.6 | 15.75 | 16.81 | 149.3 | 12.25 | 11.11 |
| | Ours | 90.7 | 24.57 | 24.35 | 116.8 | 17.74 | 17.28 | 148.0 | 12.04 | 11.71 | 177.5 | 9.26 | 8.68 |
| | Ours+LiteVAR | **110.9** | **18.92** | **18.32** | **145.4** | **12.43** | **12.03** | **172.7** | **8.85** | **8.24** | **180.6** | **9.12** | **8.48** |

we report FID2FP16 (Tang et al., 2024), which measures the FID score between outputs of full-precision VAR and its quantized counterpart. For Infinity (Han et al., 2025), we evaluate our method on text-to-image generation with ImageReward (Xu et al., 2023), HPSv2.1 (Wu et al., 2023), and GenEval (Ghosh et al., 2023), following the experimental protocols in (Han et al., 2025).

Following (Tian et al., 2024), we evaluate inpainting, outpainting, and class-conditional editing directly on images with the validation split of ImageNet. For image inpainting/outpainting, following (Zhuang et al., 2024; Suvorov et al., 2022), we measure average LPIPS (Zhang et al., 2018) between generated and original images, where the generated images are produced by filling a randomly masked region covering 25% of the image area. For class-conditional editing, we convert the ImageNet class label into a text prompt (*"a photo of [classname]"*) and compute CLIP scores (Radford et al., 2021) between the text prompt and generated image, following (Radford et al., 2021).

We built our method on top of BRECQ (Li et al., 2021) to quantize VAR, calibrating the quantization parameters using 256 images obtained either by random sampling or by our calibration data resampling method described in Sec. 3.2.2. We initialize the step size and zero-point using Percentile method (Li et al., 2019) for weight quantizers. Following LiteVAR (Xie et al., 2024), we exploit dynamic min-max quantizers for activations. Given a BOP constraint, we determine the minimum value of $\theta$ that satisfies the constraint through a grid search over $[0, 1]$ with a step size of 1e-4. We impose an additional BOP constraint equal to 1% of the total operations required for full inference, unless stated otherwise.

## 4.2 RESULTS

**Class-conditional image generation.** We show in Table 1 the quantitative comparison of our method with prior approaches for the task of class-conditional image generation on ImageNet (Deng et al., 2009). We can see that our method consistently outperforms BRECQ (Li et al., 2021) across all bit-widths and architectures, demonstrating the effectiveness of the proposed shift-and-sum quantization and calibration data resampling techniques. In addition, our method achieves quantization performance comparable to LiteVAR (Xie et al., 2024), although it retains FC layers after GELU (Hendrycks & Gimpel, 2016) non-linearity in full-precision, which is suboptimal in terms of efficiency. Finally, applying our approach in combination with LiteVAR (Xie et al., 2024) yields further improvements, suggesting that the two approaches are complementary. We visualize the generated images using each method in Appendix G.1.

**Text-to-image generation.** We provide in Table 2 the quantitative comparison of our method with state-of-the-art methods for the task of text-to-image generation. We can see that our method con-

Table 2: Quantitative comparison of quantizing Infinity-2B (Han et al., 2025) for text-to-image generation. We denote by W/A the bit-widths of weights (W) and activations (A), respectively.

| #Bits (W/A) | Methods | ImageReward ↑ | HPSv2.1 ↑ | GenEval ↑ |
|---|---|---|---|---|
| 16/16 | Full-precision | 0.932 | 32.23 | 0.736 |
| 6/6 | BRECQ (Li et al., 2021) | 0.908 | 32.01 | 0.719 |
| | LiteVAR (Xie et al., 2024) | 0.912 | 32.01 | 0.720 |
| | Ours | 0.914 | 32.03 | 0.722 |
| | Ours+LiteVAR | **0.929** | **32.04** | **0.725** |
| 4/8 | BRECQ (Li et al., 2021) | 0.845 | 31.79 | 0.712 |
| | LiteVAR (Xie et al., 2024) | 0.866 | 32.03 | 0.719 |
| | Ours | 0.861 | 31.94 | 0.716 |
| | Ours+LiteVAR | **0.880** | **32.08** | **0.722** |
| 4/6 | BRECQ (Li et al., 2021) | 0.831 | 31.62 | 0.703 |
| | LiteVAR (Xie et al., 2024) | 0.838 | 31.61 | 0.702 |
| | Ours | 0.842 | 31.70 | 0.706 |
| | Ours+LiteVAR | **0.880** | **32.04** | **0.714** |
| 4/4 | BRECQ (Li et al., 2021) | 0.346 | 27.92 | 0.563 |
| | LiteVAR (Xie et al., 2024) | 0.407 | 28.98 | 0.626 |
| | Ours | 0.575 | 29.32 | 0.653 |
| | Ours+LiteVAR | **0.748** | **29.57** | **0.672** |

Table 3: Quantitative comparison of quantizing VAR-d30 with various methods for the tasks of image inpainting, outpainting, and class-conditional editing.

| #Bits (W/A) | Methods | LPIPS ↓ (inpaint) | LPIPS ↓ (outpaint) | CLIP scores ↑ (editing) |
|---|---|---|---|---|
| 16/16 | Full-precision | 0.2827 | 0.2119 | 0.2480 |
| 6/6 | BRECQ (Li et al., 2021) | 0.2852 | 0.2144 | 0.2455 |
| | LiteVAR (Xie et al., 2024) | 0.2861 | 0.2143 | 0.2458 |
| | Ours | 0.2843 | 0.2141 | 0.2459 |
| | Ours+LiteVAR | **0.2835** | **0.2134** | **0.2465** |
| 4/4 | BRECQ (Li et al., 2021) | 0.2912 | 0.2207 | 0.2289 |
| | LiteVAR (Xie et al., 2024) | 0.2907 | 0.2208 | 0.2318 |
| | Ours | 0.2888 | 0.2181 | 0.2358 |
| | Ours+LiteVAR | **0.2879** | **0.2172** | **0.2363** |

sistently outperforms BRECQ (Li et al., 2021) and LiteVAR (Xie et al., 2024) for all benchmarks, demonstrating its efficiency. Applying our method on top of LiteVAR further improves performance; however, LiteVAR retains the FC layers after GELU (Hendrycks & Gimpel, 2016) in full precision, which introduces additional computational inefficiency. We provide the generated images for a number of text prompts in Appendix G.2.

**Image inpainting, outpainting, and class-conditional editing.** We show in Table 3 a quantitative comparison of quantizing VAR-d30 using BRECQ (Li et al., 2021), LiteVAR (Xie et al., 2024), and our method on image inpainting, outpainting, and class-conditional editing. The overall performance differences are relatively small, since the baseline already performs close to the full-precision model on these editing tasks, leaving only a narrow margin for further improvement. Within this limited range, our method consistently achieves the best LPIPS and CLIP scores across bit-widths. Moreover, as shown in Fig. D (Appendix G.3), our quantized model produces fewer artifacts and better preserves semantic content compared to BRECQ and LiteVAR.

## 4.3 DISCUSSION

**Ablation study.** We show in Table 4 an ablation analysis on different components of our method. We find that shift-and-sum quantization consistently improves performance for both VAR-d16 and VAR-d20, indicating that alleviating reconstruction errors in attention–value multiplications is critical for effective quantization. We can also see that calibration data resampling further enhances quantization performance, suggesting that discrepancies between predicted probabilities of codebook entries and their sampling frequencies in calibration data degrade quantization performance.

**Analysis on shift-and-sum quantization.** We visualize in Fig. 5(a-b) the reconstruction error of attention-value products across scales. We can see that shift-and-sum quantization effectively reduces reconstruction errors in attention-value products, especially at coarse scales where high atten-

Table 4: Ablation analysis for different components of our method. We report the results of conditional image generation on ImageNet (Deng et al., 2009) under a 4/6-bit setting. We denote by 'Shift-and-sum' and 'Resample' the shift-and-sum quantization and calibration data resampling techniques, respectively.

| Shift-and-sum | Resample | IS | FID | FID2FP16 |
|---|---|---|---|---|
| | | 145.6 | 11.16 | 10.57 |
| ✓ | | 155.4 | 10.15 | 9.55 |
| | ✓ | 152.8 | 10.19 | 9.74 |
| ✓ | ✓ | 162.1 | 9.20 | 8.44 |

(a) VAR-d16

| Shift-and-sum | Resample | IS | FID | FID2FP16 |
|---|---|---|---|---|
| | | 189.1 | 6.74 | 6.62 |
| ✓ | | 194.8 | 6.10 | 6.03 |
| | ✓ | 195.3 | 5.86 | 5.61 |
| ✓ | ✓ | 202.2 | 5.36 | 5.09 |

(b) VAR-d20

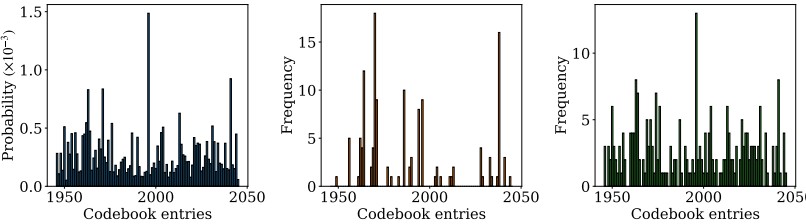

| (a) | (b) | (c) | (d) |

Figure 5: (a-b) Comparison of reconstruction error for the multiplication between attention scores and value tokens, using BRECQ (Li et al., 2021) and our method. (c-d) Visualization of IS and FID scores for VAR-d16 and VAR-d20 w.r.t. the BOP budget for conditional image generation on ImageNet (Deng et al., 2009). We visualize the reconstruction errors and performances for VAR-d16 and VAR-d20 under a 4/6-bit setting.

Figure 6: Visualization for the predicted probabilities of codebook entries (left), and their sampling frequencies before and after calibration data resampling (middle-right).

tion scores occur frequently. We show in Fig. 5(c-d) the IS and FID scores of generated images under varying BOP constraints. We observe that quantization performance consistently improves for both VAR-d16 and VAR-d20 as the BOP budget increases, indicating that higher kernel orders more effectively reduce reconstruction errors in attention–value products. We also find that the improvement saturates as the BOP constraint reaches 1% of the total inference cost, suggesting that shift-and-sum quantization alleviates reconstruction errors with only a marginal overhead.

**Analysis on calibration data resampling.** We compare in Fig. 6 the predicted probability of each codebook entry aggregated over calibration samples, and its sampling frequency before and after applying calibration data resampling technique. We can see that our resampling technique effectively aligns sampling frequencies of codebook entries with their probabilities, allowing the calibration data to better reflect the underlying probability distribution.

## 5 CONCLUSION

We have observed that applying PTQ to VAR suffers from large reconstruction errors in attention-value multiplications, which are amplified in the presence of high attention scores. To this end, we have introduced shift-and-sum quantization, which mitigate these errors by applying quantization kernels to attentive tokens. In addition, we have found that a mismatch often arises between the predicted probabilities of codebook entries and their sampling frequencies due to limited calibration samples in PTQ. Based on this, we have proposed calibration data resampling technique that aligns sampling frequencies with the underlying probability distribution. We have demonstrated the effectiveness of our method through extensive experiments on class-conditional generation, inpainting, outpainting, and class-conditional editing across VAR architectures.

## ACKNOWLEDGEMENTS

This work was supported in part by Institute of Information & Communications Technology Planning & Evaluation (IITP) grant funded by the Korea government (MSIT) (No.RS-2022-00143524, Development of Fundamental Technology and Integrated Solution for Next-Generation Automatic Artificial Intelligence System, No.RS-2025-09942968, AI Semiconductor Innovation Lab (Yonsei University)), and the National Research Foundation of Korea (NRF) grant funded by the Korea government (MSIT) (RS-2025-02216328).

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

Table A: Magnitude of pearson correlation coefficients between quantization noises. We evaluate the correlation between $\tilde{\epsilon}_i^a$ and $\epsilon_{ij}^v$ across 1,000 randomly sampled images using VAR-d30.

| Noise Pair | Pearson Correlation Coefficient |
|---|---|
| $(\tilde{\epsilon}_i^a,\ \tilde{\epsilon}_k^a)_{i \neq k}$ | $2.5 \times 10^{-6}$ |
| $(\tilde{\epsilon}_i^a,\ \epsilon_{ij}^v)$ | $1.7 \times 10^{-7}$ |
| $(\epsilon_{ij}^v,\ \epsilon_{kj}^v)_{i \neq k}$ | $2.3 \times 10^{-6}$ |

## A    CORRELATION BETWEEN QUANTIZATION NOISES

Eq. 8 assumes that the quantization errors of attention scores, and those of value tokens are independent across tokens. Although this assumption is introduced only to simplify the derivation of Eq. 8, we verify here that it closely matches actual behavior in practice.

We show in Table A the magnitude of Pearson correlation coefficients between $\tilde{\epsilon}_i^a$ and $\epsilon_{ij}^v$ on 1,000 randomly sampled images using VAR-d30. In addition, we measure the magnitudes of correlations between quantization errors across different attention rows and different value tokens. We can see that all measured correlations are extremely small (typically an order of $10^{-6}$), indicating that the error terms are uncorrelated in practice.

## B    PROOF OF EQ. 8

We obtain the reconstruction error in Eq. 7 for the multiplication between attention scores $\mathbf{a} \in \mathbb{R}^T$ and value tokens $V \in \mathbb{R}^{T \times d}$ as follows. For the $j$-th dimension, the reconstruction error is

$$\delta_j = \sum_{i=1}^{T} a_i \big( \tilde{\epsilon}_i^a v_{ij} + \tilde{\epsilon}_i^a \epsilon_{ij}^v + \epsilon_{ij}^v \big), \tag{21}$$

where we denote $v_{ij} \in \mathbb{R}$ and $\epsilon_{ij}^v \in \mathbb{R}$ as the $j$-th component of the $i$-th value token, and the corresponding quantization noise.

Since the quantization noises $\tilde{\epsilon}_i^a$ and $\epsilon_{ij}^v$ are independent across tokens, the variance of $\delta_j$ can be written as:

$$\mathrm{Var}[\delta_j] = \mathrm{Var}\Big[ \sum_{i=1}^{T} a_i \big( \tilde{\epsilon}_i^a v_{ij} + \tilde{\epsilon}_i^a \epsilon_{ij}^v + \epsilon_{ij}^v \big) \Big] = \sum_{i=1}^{T} a_i^2 \mathrm{Var}[\tilde{\epsilon}_i^a v_{ij} + \tilde{\epsilon}_i^a \epsilon_{ij}^v + \epsilon_{ij}^v]. \tag{22}$$

We can expand the variance term as follows:

$$\mathrm{Var}[\tilde{\epsilon}_i^a v_{ij} + \tilde{\epsilon}_i^a \epsilon_{ij}^v + \epsilon_{ij}^v] = \mathrm{Var}[\tilde{\epsilon}_i^a v_{ij}] + \mathrm{Var}[\tilde{\epsilon}_i^a \epsilon_{ij}^v] + \mathrm{Var}[\epsilon_{ij}^v] \\ + 2\mathrm{Cov}[\tilde{\epsilon}_i^a v_{ij}, \tilde{\epsilon}_i^a \epsilon_{ij}^v] + 2\mathrm{Cov}[\tilde{\epsilon}_i^a v_{ij}, \epsilon_{ij}^v] + 2\mathrm{Cov}[\tilde{\epsilon}_i^a \epsilon_{ij}^v, \epsilon_{ij}^v]. \tag{23}$$

Based on the assumption that $\{\tilde{\epsilon}_i^a\}_{i=1}^T$ and $\{\epsilon_{ij}^v\}_{i=1}^T$ are independent zero-mean noises, the covariance terms vanish as follows:

$$\mathrm{Cov}[\tilde{\epsilon}_i^a v_{ij}, \tilde{\epsilon}_i^a \epsilon_{ij}^v] = v_{ij}\, \mathbb{E}[(\tilde{\epsilon}_i^a)^2 \epsilon_{ij}^v] - v_{ij}\, \mathbb{E}[\tilde{\epsilon}_i^a]\mathbb{E}[\tilde{\epsilon}_i^a \epsilon_{ij}^v] = 0, \tag{24}$$

$$\mathrm{Cov}[\tilde{\epsilon}_i^a v_{ij}, \epsilon_{ij}^v] = v_{ij}\mathrm{Cov}[\tilde{\epsilon}_i^a, \epsilon_{ij}^v] = 0, \tag{25}$$

$$\mathrm{Cov}[\tilde{\epsilon}_i^a \epsilon_{ij}^v, \epsilon_{ij}^v] = \mathbb{E}[\tilde{\epsilon}_i^a (\epsilon_{ij}^v)^2] - \mathbb{E}[\tilde{\epsilon}_i^a \epsilon_{ij}^v]\mathbb{E}[\epsilon_{ij}^v] = 0. \tag{26}$$

Thus Eq. 23 simplifies to

$$\mathrm{Var}[\tilde{\epsilon}_i^a v_{ij} + \tilde{\epsilon}_i^a \epsilon_{ij}^v + \epsilon_{ij}^v] = \sigma_a^2 v_{ij}^2 + \sigma_a^2 \sigma_v^2 + \sigma_v^2. \tag{27}$$

Substituting into Eq. 22, we obtain

$$\mathrm{Var}[\delta_j] = \sum_{i=1}^{T} a_i^2 \big( \sigma_a^2 v_{ij}^2 + \sigma_a^2 \sigma_v^2 + \sigma_v^2 \big). \tag{28}$$

Finally, summing across dimensions $j = 1, \ldots, d$ gives the total variance

$$\text{Var}[\delta] = \sum_{j=1}^{d} \text{Var}[\delta_j] = \sum_{i=1}^{T} a_i^2 \Big( \sigma_a^2 \sum_{j=1}^{d} v_{ij}^2 + d(\sigma_a^2 \sigma_v^2 + \sigma_v^2) \Big), \tag{29}$$

which corresponds to Eq. 8 and completes the proof.

## C  PROOF OF THEOREM 1

We will prove the following in sequence: **(i)** using $t_n = s/4n$ provides an upper bound of $s/4n$, and **(ii)** no upper bound smaller than $s/4n$ can be obtained for arbitrary shift factor $t_n$. Note that we omit zero-points for clarity.

**(i).** Suppose we set $t_n = s/4n$. In this case, we will first show that the quantization kernel in Eq. 9 reduces to $\frac{s}{2n} \lfloor \frac{2nv}{s} \rceil$. We define

$$\begin{aligned}
g(v) &= \frac{s}{2n} \lfloor \frac{2nv}{s} \rceil - f_n(v; \frac{s}{4n}) \\
&= \frac{s}{2n} \lfloor \frac{2nv}{s} \rceil - \frac{1}{2n} \sum_{k=-n}^{n-1} Q\Big( v + (2k+1)\frac{s}{4n} \Big).
\end{aligned} \tag{30}$$

Adding $s/2n$ to $v$, we can obtain:

$$g\Big( v + \frac{s}{2n} \Big) = \frac{s}{2n} \lfloor \frac{2nv}{s} + 1 \rceil - \frac{1}{2n} \sum_{k=-n}^{n-1} Q\Big( v + (2k+3)\frac{s}{4n} \Big). \tag{31}$$

We can see that $g(v)$ is periodic as follows:

$$\begin{aligned}
g\Big( v + \frac{s}{2n} \Big) &= \frac{s}{2n} \lfloor \frac{2nv}{s} + 1 \rceil - \frac{1}{2n} \sum_{k=-n}^{n-1} Q\Big( v + (2k+3)\frac{s}{4n} \Big) \\
&= \frac{s}{2n} \lfloor \frac{2nv}{s} \rceil + \frac{s}{2n} - \frac{1}{2n} \left( \sum_{k=-n}^{n-1} Q_v\Big( v + (2k+1)\frac{s}{4n} \Big) + s \right) \\
&= g(v).
\end{aligned} \tag{32}$$

Furthermore, $g(v) = 0$ for $v \in [-s/4n, \; s/4n)$; by periodicity this implies $g(v) = 0$ for all $v$. Hence the quantization error of $f_n(v; s/4n)$ is bounded as

$$\begin{aligned}
\Big| v - f_n(v; \frac{s}{4n}) \Big| &= \Big| v - \frac{s}{2n} \Big\lfloor \frac{2nv}{s} \Big\rceil \Big| \\
&= \frac{1}{2n} \Big| 2nv - s \Big\lfloor \frac{2nv}{s} \Big\rceil \Big| \\
&\leq \frac{s}{4n},
\end{aligned} \tag{33}$$

which establishes part **(i)**.

**(ii).** We now prove tightness of the bound. Suppose, for contradiction, that there exists a shift factor $t_n$ such that

$$\max_v |v - f_n(v; t_n)| < \frac{s}{4n}. \tag{34}$$

Since $Q(\cdot)$ outputs values on the quantization grid $\{ms : m \in \mathbb{Z}\}$, the averaging operation in $f_n(v; t_n)$ produces outputs confined to the finer grid

$$\Big\{ \frac{ms}{2n} : m \in \mathbb{Z} \Big\}. \tag{35}$$

Table B: Analysis of computational overheads (i.e., memory, BOPs) introduced by shift-and-sum quantization, and quantization performances of VAR-d16 under a 4/6-bit setting. Note that we do not use calibration data resampling in this analysis.

|  | $\theta = 0.05$ | | $\theta = 0.1$ | | $\theta = 0.2$ | |
|---|---|---|---|---|---|---|
|  | Memory | BOPs | Memory | BOPs | Memory | BOPs |
| Baseline | 1.03G | 3.72T | 1.03G | 3.72T | 1.03G | 3.72T |
| Eq. 11 | 0.22M | 1.17G | 0.22M | 1.17G | 0.22M | 1.17G |
| Duplicate & Shift | 23.21M | 0.28G | 9.58M | 0.13G | 3.94M | 0.06G |
| Eq. 13 | – | 33.35G | – | 11.19G | – | 3.94G |
| Total | 1.05G | 3.75T | 1.04G | 3.73T | 1.03G | 3.72T |

**(a)** Computational overheads

| Method | IS | FID | FID2FP16 |
|---|---|---|---|
| Baseline | 145.6 | 11.16 | 10.57 |
| Shift-and-sum ($\theta = 0.05$) | 155.4 | 10.15 | 9.55 |
| Shift-and-sum ($\theta = 0.1$) | 155.2 | 10.25 | 9.57 |
| Shift-and-sum ($\theta = 0.2$) | 154.3 | 10.33 | 9.71 |

**(b)** Quantization performances

Now suppose we set $v$ to $s/4n$. For this choice, we obtain

$$\left| \frac{s}{4n} - f_n(\frac{s}{4n}; t_n) \right| = \left| \frac{s}{4n} - \frac{\hat{m}s}{2n} \right| \geq \frac{s}{4n}, \tag{36}$$

where $\hat{m}$ is an integer. Therefore, we conclude that

$$\max_v |v - f_n(v; t_n)| \geq \frac{s}{4n}, \tag{37}$$

contradicting Eq. 34. Hence no choice of $t_n$ can achieve a bound smaller than $s/4n$, which establishes part **(ii)**.

## D    COMPUTATIONAL OVERHEAD

Compared to baseline, shift-and-sum quantization introduces additional memory usage and BOP overhead during inference. We can see from Fig. 3 that shift-and-sum quantization requires (i) computing average attention scores for each value token (Eq. 11), (ii) duplicating and shifting tokens with high attention scores, and (iii) aggregating the quantized results from their symmetrically shifted counterparts (Eq. 13).

Consider a matrix multiplication between attention matrix $\mathbf{A} \in \mathbb{R}^{T' \times T}$ and value matrix $\mathbf{V} \in \mathbb{R}^{T \times d}$, where we apply a quantization kernel with an order of $n$ to a single token in $\mathbf{V}$. Computing the average attention scores (Eq. 11) requires additional $16T$ bits for storage and $16T'T$ BOPs. Duplicating and shifting attentive tokens, together with their corresponding attention scores, introduces additional $16(2n - 1)(T' + d)$ extra bits and $2n(T' + 16d)$ BOPs. Finally, aggregating the quantized outputs (Eq. 13) incurs $(2n - 1)b_x^2 dT'$ additional BOPs, where we denote $b_x$ as the bit-widths of activations.

In practice, shift-and-sum quantization is applied only to tokens whose average attention scores exceed the threshold $\theta$, and the order $n$ remains small for most tokens. Consequently, the computational overhead for shift-and-sum quantization is relatively marginal compared to the whole inference process. We report in Table B(a) the memory and BOP overheads under a 4/6-bit setting for VAR-d16 with varying threshold $\theta$. We show in Table B(b) the quantization performances of VAR-d16 for varying $\theta$. We can see that our method improves all three metrics (IS, FID, and FID2FP16) consistently, while introducing only minor computational overhead.

## E    THROUGHPUT

We compare in Table C the throughputs of VAR-d16 under different quantization methods, where we convert the precision of weights and activations into 8 bits using ONNX Runtime. All quantized

Table C: Throughputs (image/sec) of VAR-d16 under different quantization methods, measured using ONNX Runtime on an Intel Xeon Gold 6226R CPU.

| Method | Bits (W/A) | Throughput (image/sec) |
|---|---|---|
| Full-precision | 16/16 | $0.1208 \pm 0.0010$ |
| BRECQ (Li et al., 2021) | 8/8 | $0.2195 \pm 0.0046$ |
| LiteVAR (Xie et al., 2024) | 8/8 | $0.1749 \pm 0.0028$ |
| Ours | 8/8 | $0.2147 \pm 0.0042$ |
| Ours+LiteVAR | 8/8 | $0.1719 \pm 0.0026$ |

Table D: Quantitative comparison of quantizing OneFormer (Jain et al., 2023) with various quantization methods for the tasks of semantic, instance, and panoptic segmentation on COCO.

| Bits (W/A) | Method | mIoU | AP | PQ |
|---|---|---|---|---|
| 16/16 | Full-precision | 67.4 | 49.0 | 57.9 |
| 6/6 | BRECQ (Li et al., 2021) | 66.5 | 47.8 | 55.8 |
| | LiteVAR (Xie et al., 2024) | 66.9 | 48.2 | 57.4 |
| | Ours | 66.8 | 48.3 | 57.6 |
| | Ours+LiteVAR | **67.0** | **48.6** | **57.8** |
| 4/4 | BRECQ (Li et al., 2021) | 63.7 | 37.9 | 51.0 |
| | LiteVAR (Xie et al., 2024) | 64.4 | 39.2 | 51.5 |
| | Ours | 64.6 | 39.1 | 51.6 |
| | Ours+LiteVAR | **65.0** | **40.6** | **52.2** |

models are exported to the ONNX format and executed with 8-bit integer operations on an Intel Xeon Gold 6226R CPU, allowing us to report actual images-per-second throughput rather than estimates based on simulated quantization in floating-point arithmetic. We also include a variant where the FC layer after GELU non-linearity is kept in full precision, following the quantization strategy adopted in LiteVAR (Xie et al., 2024). We can see that our method is only about 2% slower than BRECQ (Li et al., 2021), while being considerably faster than LiteVAR, highlighting the practical efficiency of our approach for real deployment scenarios.

## F  APPLICATION TO ONEFORMER

We show in Table D the results for quantizing OneFormer (Jain et al., 2023) for the tasks of semantic, instance, and panoptic segmentation on COCO (Lin et al., 2014). We measure mIoU (Everingham et al., 2015), AP (Lin et al., 2014), and PQ (Kirillov et al., 2019) for semantic, instance, and panoptic segmentation, respectively. We observe that our method outperforms BRECQ (Li et al., 2021) consistently, while having comparable performance with LiteVAR (Xie et al., 2024), despite LiteVAR keeping the FC layers after the GELU (Hendrycks & Gimpel, 2016) non-linearity in full precision, leading to computational inefficiency. Finally, combining our method with LiteVAR yields further improvements across all three tasks.

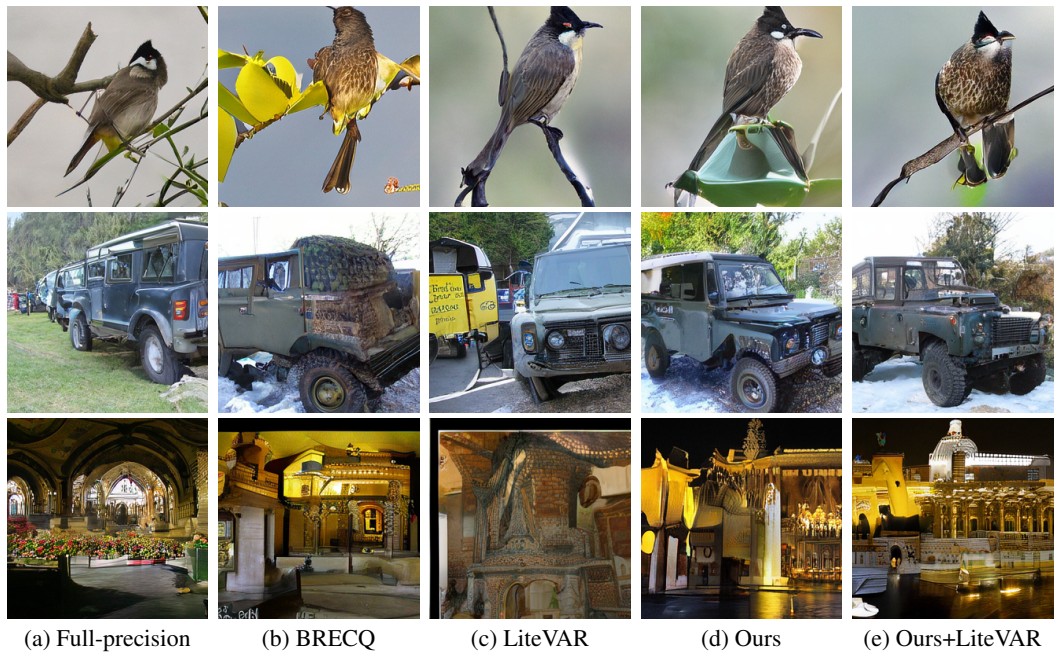

| (a) Full-precision | (b) BRECQ | (c) LiteVAR | (d) Ours | (e) Ours+LiteVAR |

Figure A: Visualization of generated images for a full-precision VAR-d16 and its counterparts quantized with BRECQ (Li et al., 2021), LiteVAR (Xie et al., 2024), and our method under a 6/6-bit setting.

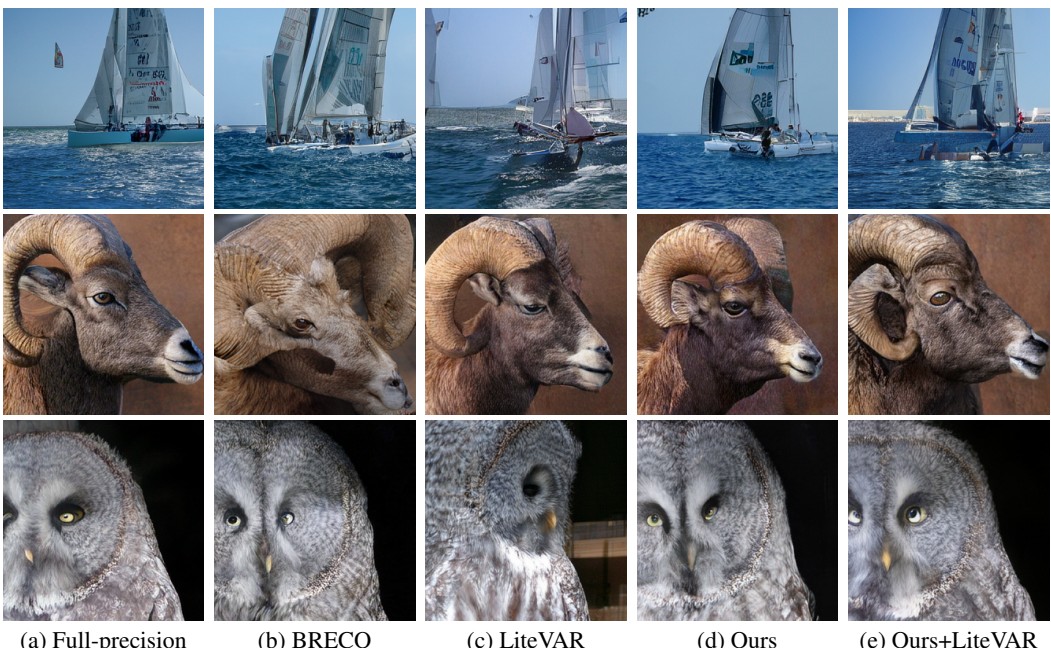

| (a) Full-precision | (b) BRECQ | (c) LiteVAR | (d) Ours | (e) Ours+LiteVAR |

Figure B: Visualization of generated images for a full-precision VAR-d16 and its counterparts quantized with BRECQ (Li et al., 2021), LiteVAR (Xie et al., 2024), and our method under a 6/6-bit setting. Note that we teacher-force the token maps sampled from the full-precision model for the first two stages in VAR, unlike Fig. A.

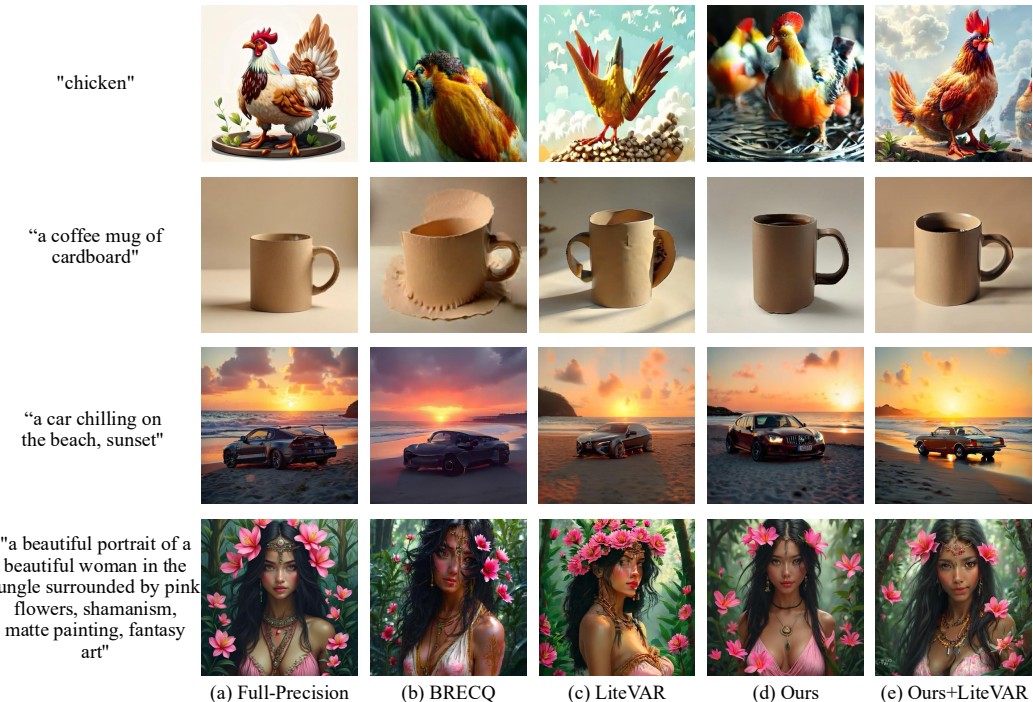

Figure C: Comparison of generated images using Infinity-2B (Han et al., 2025) and its quantized counterparts using different methods.

# G  QUALITATIVE RESULTS

## G.1  CONDITIONAL IMAGE GENERATION

We show in Fig. A a qualitative comparison of images generated by VAR-d16 quantized with BRECQ (Li et al., 2021), LiteVAR (Xie et al., 2024), and our method. For each row, we present results generated from the same random seed and ImageNet label to enable direct visual comparison across methods.

Since VAR samples token maps from the predicted probability distribution at each stage, even slight perturbations at coarse scales can cause different codebook entries to be chosen, resulting in drastically different output images. To address this, we additionally report results in Fig. B, where we apply teacher-forcing to align the sampled codebook entries with the full-precision model in early stages, while allowing later stages to proceed autoregressively. This ensures consistent global structure across methods and highlights the impact of different quantization methods on fine-scale details. We can see that our method generates images that are more realistic and closer to those of the full-precision model compared to other approaches.

## G.2  TEXT-TO-IMAGE GENERATION

We show in Fig. C a qualitative comparison of images generated by Infinity-2B quantized with BRECQ (Li et al., 2021), LiteVAR (Xie et al., 2024), and our method. We can see that our method consistently produces images with better fidelity compared to other methods.

## G.3  IMAGE INPAINTING, OUTPAINTING, AND CLASS-CONDITIONAL EDITING

We present in Fig. D a qualitative comparison of a VAR-d30 model quantized using BRECQ (Li et al., 2021), LiteVAR (Xie et al., 2024), and our method on the tasks of inpainting (1st–3rd rows), outpainting (4th–6th rows), and class-conditional editing (7th–8th rows). We observe that our method consistently produces better global structure and fewer artifacts compared to other methods.

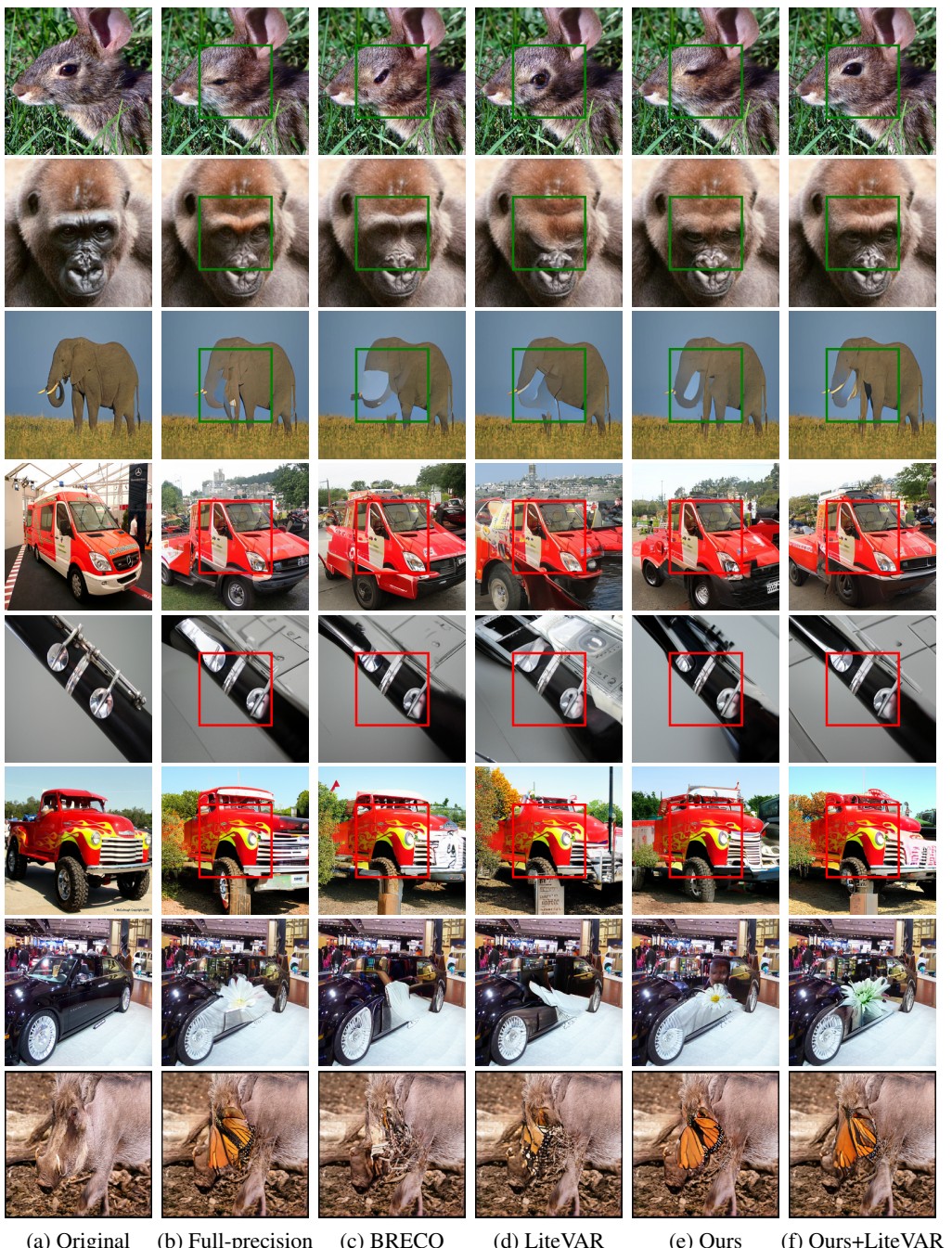

    (a) Original    (b) Full-precision    (c) BRECQ    (d) LiteVAR    (e) Ours    (f) Ours+LiteVAR

Figure D: Qualitative comparison of results for the tasks of image inpainting (1st–3rd rows), outpainting (4th–6th rows), and class-conditional editing (7th–8th rows) using VAR-d30.

## H   BIT-WIDTH VS. PERFORMANCE

We provide in Fig. E qualitative comparisons of images generated by VAR-d30 under different bit-width settings using our method. The results show that generation quality remains largely stable at 6/6- and 4/8-bit configurations, while a significant degradation appears at the more aggressive 4/4-bit setting. This visualization highlights how our method performs under varying compression levels and directly illustrates the trade-off between bit-width and perceptual quality.

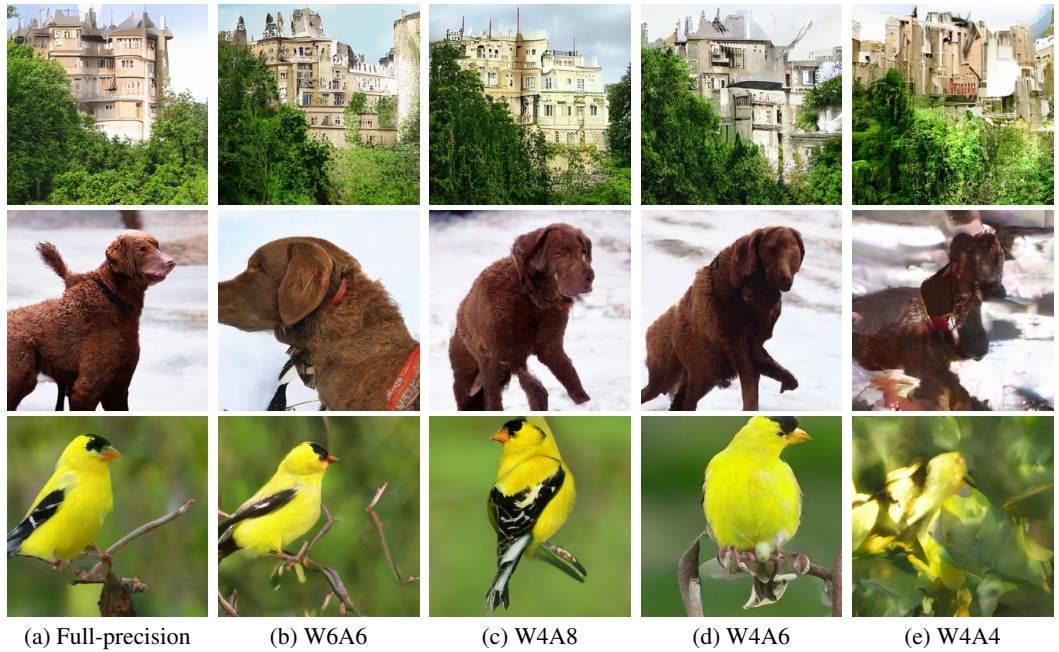

|  (a) Full-precision | (b) W6A6 | (c) W4A8 | (d) W4A6 | (e) W4A4 |

Figure E: Visualization of images generated by a full-precision VAR-d30 and its quantized counterparts under different bit-width settings using our method.

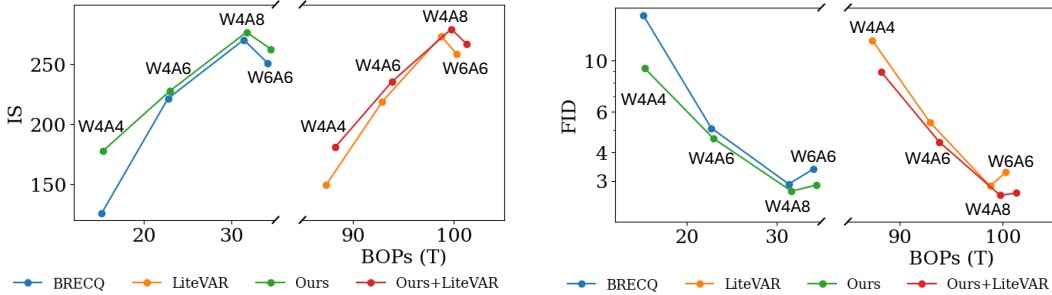

Figure F: Visualization of IS and FID scores for VAR-d30 w.r.t. the BOP budget for conditional image generation on ImageNet (Deng et al., 2009).

In addition, we present in Fig. F the IS and FID scores of VAR-d30 quantized with various bit-width settings. Our method consistently provides a more favorable quality–efficiency trade-off compared to BRECQ (Li et al., 2021) and LiteVAR (Xie et al., 2024). For settings with tight BOP budgets, our approach yields noticeably larger performance gains—achieving higher IS and lower FID than both methods—which indicates that the proposed method preserves generation quality substantially better when computation is highly constrained. This demonstrates that our method is more robust to quantization and maintains higher fidelity across a wide range of compression levels, thereby offering a superior balance between computational cost and generation quality.

We also observe that in some regions of the trade-off curves, a configuration with a lower BOP budget can unexpectedly outperform one with a higher budget (e.g., W4A8 and W6A6). This occurs because activation quantization is typically more critical to the final generation quality than weight quantization. As a result, a configuration with fewer operations but higher-precision activations can sometimes outperform another with more operations but lower-precision activations.

## I   LIMITATIONS & FUTURE WORK

Although we demonstrate the applicability of our method beyond VAR using OneFormer (Jain et al., 2023), evaluating a broader range of transformer architectures remains an interesting direction for future work. In addition, the effectiveness of our method degrades when the precision is pushed below 4 bits. In such extremely low-precision regimes, quantization noise could dominate the generation process, leading to severe performance degradation across all evaluated methods. Addressing these regimes likely requires fundamentally different quantization strategies or training-aware approaches, which we leave for future work.

## J   LARGE LANGUAGE MODEL (LLM) USAGE

In this paper, we use LLMs solely to aid in polishing the writing. No LLMs were used for ideation, retrieval of related work, or experimental design.

