# OpenReview forum: "Shift-and-Sum Quantization for Visual Autoregressive Models"
_ICLR.cc/2026/Conference — ICLR 2026 Poster_

### Official Review · Reviewer_ND1G · 2025-10-27

**Soundness:** 3
**Presentation:** 3
**Contribution:** 3
**Rating:** 6
**Confidence:** 3

**Summary:**

This paper applied post-training quantization (PTQ) for visual autoregressive models (VAR), including a shift-and-sum quantization method to reduce calibration data and a resampling strategy for calibration data to align sampling frequencies of codebook entries with their predicted probabilities.

**Strengths:**

- The paper fills a the gap by explicitly identifying VAR-specific quantization challenges, i.e., the attention-value error amplification at coarse scales and the codebook frequency-probability mismatch.
- The theoretical analysis is sufficient, e.g., the theoretical analysis (Theorem 1) that proves the error bound for the proposed shift-and-sum quantization.

**Weaknesses:**

- Insufficient Analysis of Computational Overhead: The proposed shift-and-sum quantization introduces additional operations, such as shift, duplication, and aggregation, which may increase inference time and memory consumption. However, the paper lacks a thorough analysis or empirical evaluation of these computational costs. A detailed study on the overhead introduced by these operations is necessary to fully assess the practicality of the method.
- Qualitative Results Show Noticeable Degradation: The qualitative results presented demonstrate a clear degradation compared to full-precision models. To better illustrate the trade-off between compression rate and generation quality, the authors should provide a comprehensive comparison of generated results across different bit-widths. Additionally, it would be beneficial to include trade-off curves comparing the proposed method with other baseline approaches.
- Limited Evaluation Metrics: The paper primarily adopts FID and IS as evaluation metrics, which mainly assess the generation quality for inpainting and outpainting tasks. However, these metrics do not adequately capture the semantic alignment between the generated results and the conditional guidance. The authors should consider incorporating additional metrics or evaluation protocols to assess semantic consistency and alignment.

**Questions:**

Please refer to the weaknesses.

---

> ### Author Response · Authors · 2025-11-24
> **Response to Reviewer ND1G [1/2]**
>
> We sincerely appreciate the constructive comments provided by Reviewer ND1G. Our detailed responses to the comments are presented below.
> *****
> > **W1. Insufficient Analysis of Computational Overhead**
>
> We agree that understanding the cost introduced by token duplication, shifting, and aggregation is essential for assessing the practicality of the method. To address this, we have added a detailed analysis in **Table R1**, which is also included in Table B(a) of the revised paper. It provides:
> - a breakdown of the additional memory usage and BOPs (bit-operations) associated with each component of shift-and-sum quantization,
> - and an empirical comparison of overheads under different thresholds θ∈{0.05,0.1,0.2}, which determine how many tokens are processed with higher-order kernels.
>
> In addition, we report in **Table R2** (included in Table B(b) of the revised paper) the corresponding quantization performance (e.g., IS, FID, FID2FP16) for each θ value. Our results show that the computational overhead remains marginal (less than 1% of total BOPs) for practical θ values, while offering clear improvements in quantization performance. We hope that this analysis clarifies the practicality of our method and addresses the reviewer’s concern.
>
> **Table R1**: Analysis of computational overheads (i.e., memory, BOPs) introduced by shift-and-sum quantization (4/6-bit setting).
> | Method                | Memory (θ=0.05) | BOPs (θ=0.05) | Memory (θ=0.1) | BOPs (θ=0.1) | Memory (θ=0.2) | BOPs (θ=0.2) |
> |-----------------------|------------------|----------------|-----------------|----------------|-----------------|----------------|
> | Baseline              | 1.03G            | 3.72T          | 1.03G           | 3.72T          | 1.03G           | 3.72T          |
> | Eq. (11)   | 0.22M            | 1.17G          | 0.22M           | 1.17G          | 0.22M           | 1.17G          |
> | Duplicate & Shift     | 23.21M           | 0.28G          | 9.58M           | 0.13G          | 3.94M           | 0.06G          |
> | Eq. (13)        | --               | 33.35G         | --              | 11.19G         | --              | 3.94G          |
> | **Total**             | **1.05G**        | **3.75T**      | **1.04G**       | **3.73T**      | **1.03G**       | **3.72T**      |
>
>
> **Table R2**: Quantization performances of VAR-d16 for varying θ (4/6-bit setting).
> | Method                     | IS    | FID   | FID2FP16 |
> |----------------------------|-------|-------|-----------|
> | Baseline                   | 145.6 | 11.16 | 10.57     |
> | Shift-and-sum (θ = 0.05)   | 155.4 | 10.15 | 9.55      |
> | Shift-and-sum (θ = 0.10)   | 155.2 | 10.25 | 9.57      |
> | Shift-and-sum (θ = 0.20)   | 154.3 | 10.33 | 9.71      |
> *****
> > **W2. Qualitative Results Show Noticeable Degradation & Tradeoff curves**
>
> We thank the reviewer for highlighting the need for a visualization on various bit-widths, and a more comprehensive trade-off analysis. We have shown the qualitative results for various bit-width settings in Fig. E of the revised paper. In addition, we have provided in Fig. F the IS–BOP and FID–BOP trade-off curves for VAR-d30. These results clearly show that our method achieves a more favorable quality–efficiency tradeoff compared to both BRECQ and LiteVAR.

---

> ### Author Response · Authors · 2025-11-24
> **Response to Reviewer ND1G [2/2]**
>
> > **W3. Limited Evaluation Metrics**
>
> We appreciate the reviewer’s suggestion to incorporate metrics that capture semantic alignment. In the revised paper, we now evaluate (1) LPIPS for image inpainting and outpainting—following widely adopted protocols in [C1, C2]—and (2) CLIP scores for class-conditional editing, using the standard prompt template “*a photo of [class name]*” following [C3, C4]. These metrics directly quantify perceptual quality and semantic alignment compared to IS/FID.
>
> We present in **Table R3** (included as Table 3 in the revised paper) the LPIPS and CLIP scores across different methods, showing that our method consistently achieves lower LPIPS and higher CLIP scores compared to BRECQ and LiteVAR, demonstrating improved semantic consistency after quantization.
>
> [C1] Resolution-Robust Large Mask Inpainting With Fourier Convolutions, WACV 2022\
> [C2] A Task Is Worth One Word: Learning with Task Prompts for High-Quality Versatile Image Inpainting, ECCV 2024\
> [C3] Learning transferable visual models from natural language supervision, ICML 2021\
> [C4] LEDITS++: Limitless Image Editing using Text-to-Image Models, CVPR 2024
>
> **Table R3**: Quantitative comparison of quantizing VAR-d30 with various methods for the tasks of image inpainting, outpainting, and class-conditional editing.
>
> | Bits (W/A) | Method        | LPIPS ↓ (inpaint) | LPIPS ↓ (outpaint) | CLIP score ↑ (editing) |
> |------------|---------------|-------------------|--------------------|-------------------------|
> | 16/16      | Full-precision | 0.2827            | 0.2119             | 0.2480                  |
> | 6/6        | BRECQ          | 0.2852            | 0.2144             | 0.2455                  |
> | 6/6        | LiteVAR        | 0.2861            | 0.2143             | 0.2458                  |
> | 6/6        | Ours           | 0.2843            | 0.2141             | 0.2459                  |
> | 6/6        | **Ours+LiteVAR** | **0.2835**       | **0.2134**         | **0.2465**              |
> | 4/4        | BRECQ          | 0.2912            | 0.2207             | 0.2289                  |
> | 4/4        | LiteVAR        | 0.2907            | 0.2208             | 0.2318                  |
> | 4/4        | Ours           | 0.2888            | 0.2181             | 0.2358                  |
> | 4/4        | **Ours+LiteVAR** | **0.2879**       | **0.2172**         | **0.2363**              |

---

### Official Review · Reviewer_TesS · 2025-10-28

**Soundness:** 2
**Presentation:** 3
**Contribution:** 2
**Rating:** 6
**Confidence:** 2

**Summary:**

This work analyzes the post training quantization of VAR models and points out two VAR problems: 1. significant quantization errors from the multiplication between attention scores and value tokens. 2. a mismatch between the predicted probabilities over the entries of VQVAE codebook and their sampling frequencies during calibration (Line 71). The authors propose shift-and-sum quantization which can reduce error with $O(s/4n)$ bound (Theorem 1), and calibration data resample that can resolve the mismatch. The author also provide empirical validations to show that the method improves over BRECQ and is competitive with LiteVAR under W/A bit-widths.

**Strengths:**

$\bullet$ Theoretical Contribution: this work formalizes coarse-scale attention make post-training quantization error worse, and the authors derived a variance expression (Eq. 8).

$\bullet$ shift-and-sum kernel has tight error bound: the error bound is tight $|v-f_n(v;t_n) | \leq s/(4n)$

$\bullet$ the calibration fix is simple but effective: I think the resampling technique (probablity-matching) is easy to add to the top of existing post training quantization pipeline.

$\bullet$ Experiments are comprehensive: the authors conduct experiments on multiple VAR depths, multiple bit-settings, standard metrics (IS/FID/etc.), and qualitative tasks (in/out- painting, editing)

**Weaknesses:**

Eq. 8 relies on an unrealistic assumption:  $\\{ \tilde{\epsilon\_i^a} \\}\_{i=1}^T$ and $\\{ \mathbf{\epsilon}\_i^v \\}\_{i=1}^T$ are independent, zero-mean random variables with variances $\sigma_a^2$ and $\sigma_v^2$ respectively. I checked the proof of Eq. 8 briefly, and I found the assumption is used at Eq. 22, where $\\mathrm{Var}[ \\sum\_{i} a\_i X\_i ] = \\sum\_{i} a\_i^2 \mathrm{Var}[X\_i] $ (covariance set to 0). If the assumption is dropped, it will not get the same closed form as introducing the covariance term. Furthermore, this assumption seems unrealistic to me, and one proof to break this assumption can be the following: We define $a_t$ and $\\hat{a}_t$ as the exact attention score and quantized attention score at timestep t respectively. Then, the quantization error is defined as $e_t := a_t-\\hat{a}_t$. By the definition of Softmax, $\\sum_i {a_i} = 1$ and $\\sum_i {\\hat{a}_i} = 1$ (softmax of quantized logits), thus we have $\\sum_i {e_i} = \\sum_i {a_i}  - \\sum_i {\\hat{a}_i}  = 0$. We suppose $\\{ e_1, \\dots, e_t \\}$ are independent and at least one had nonzero variance. Then we have $\\mathrm{Var}(\\sum_i e_i) = \\sum_i \\mathrm{Var}(e_i) > 0$. Since $\\sum_i {e_i}  = 0$, we have $\\mathrm{Var}(\\sum_i e_i)  = 0$. This leads to a contradiction. This proof has shown the independence assumption is unrealistic.

**Questions:**

$\bullet$ Can authors explain the practical validity of the assumption used in Eq. 8?

$\bullet$ If the Eq. 8 assumption is removed, does it affect the main result, or does it merely complicate the derivation of the bounds?

---

> ### Author Response · Authors · 2025-11-24
> **Response to Reviewer TesS [1/2]**
>
> We sincerely appreciate the constructive comments provided by Reviewer TesS. Our detailed responses to the comments are presented below.
> *****
> > **W1. Assumption on the independence of $\tilde{\epsilon}^a_i$ and $\epsilon^v_{ij}$**
>
> We thank the reviewer for carefully examining the derivation of Eq. (8) and for raising this point.
>
> The reviewer argues that a contradiction arises because the quantized attention scores satisfy $\sum_i\hat{a}_i=1$, which implies:
>
> $$
> \sum_i e_i=\sum_i(\hat{a}_i-a_i)=0.
> $$
>
> However, this condition does not hold in our setting. In our implementation—consistent with conventional practice—we first apply a softmax operation to the logits and **then** quantize the resulting attention scores. Quantizing the softmax outputs changes their values independently, so the resulting attention scores no longer sum to 1. Thus, we can see that $\sum_i\hat{a}_i\neq1$ and $\sum_ie_i\neq0$. Accordingly, the contradiction in the reviewer’s sketch does not arise.
>
> In practice, the main role of Eq. (8) is to formalize how attentive tokens amplify the reconstruction errors in attention-value products. More detailed discussions of the practical validity of this assumption and the effect of removing it are provided in our responses to Q1 and Q2, respectively.
> *****
> > **Q1. Practical validity of the assumption in Eq. (8)**
>
> We agree that the independence assumption does not strictly hold in general. To evaluate how realistic this assumption is in practice, we empirically measured the magnitude of Pearson correlation coefficient between $\tilde{\epsilon}^a_i$ and $\epsilon^v_{ij}$ on VAR-d30. We can see from **Table R1** (also shown in Table A in the revised paper) that the empirical correlations are consistently very close to zero, suggesting that the quantization noises of attention scores and value tokens behave as approximately uncorrelated in practice. This indicates that the independence assumption serves as a reasonable approximation for deriving Eq. (8).
>
> **Table R1**: Magnitude of Pearson correlation coefficients between quantization noises.
>
> | Noise Pair                                   | Pearson Correlation Coefficient |
> |----------------------------------------------|---------------------------------|
> | $(\tilde{\epsilon}^a_i,\ \tilde{\epsilon}^a_k)_{i \neq k}$ | $2.5 \times 10^{-6}$            |
> | $(\tilde{\epsilon}^a_i,\ \epsilon^v_{ij})$               | $1.7 \times 10^{-7}$            |
> | $(\epsilon^v_{ij},\ \epsilon^v_{kj})_{i \neq k}$         | $2.3 \times 10^{-6}$            |

---

> ### Author Response · Authors · 2025-11-24
> **Response to Reviewer TesS [2/2]**
>
> > **Q2. Effect of removing the assumption in Eq. (8)**
>
> If we do not assume independence between $\{\tilde{\epsilon}^a_i\}$ and $\{\epsilon^v_{ij}\}$, the covariance terms in Eq. (23) no longer vanish and Eq. (8) does not hold. In this case, starting from Eq. (22), the variance of the reconstruction error can still be written as:
>
> $$
> \mathrm{Var}[\delta_j]
>    = \mathrm{Var}\Big[\sum_{i=1}^{T} a_i(\tilde{\epsilon}^a_i v_{ij} + \tilde{\epsilon}^a_i \epsilon^v_{ij} + \epsilon^v_{ij})\Big]
> $$
>
> $$
> =\sum_{i=1}^{T} \mathrm{Var}\Big[a_i(\tilde{\epsilon}^a_i v_{ij} + \tilde{\epsilon}^a_i \epsilon^v_{ij} + \epsilon^v_{ij})\Big] + \sum_{i\neq k}\mathrm{Cov}[a_i(\tilde{\epsilon}^a_i v_{ij} + \tilde{\epsilon}^a_i \epsilon^v_{ij} + \epsilon^v_{ij}), a_k(\tilde{\epsilon}^a_k v_{kj} + \tilde{\epsilon}^a_k \epsilon^v_{kj} + \epsilon^v_{kj})]
> $$
>
> $$
> =\sum_{i=1}^{T} a_i^2\mathrm{Var}\Big[\tilde{\epsilon}^a_i v_{ij} + \tilde{\epsilon}^a_i \epsilon^v_{ij} + \epsilon^v_{ij}\Big] + \sum_{i\neq k}a_ia_k\mathrm{Cov}[\tilde{\epsilon}^a_i v_{ij} + \tilde{\epsilon}^a_i \epsilon^v_{ij} + \epsilon^v_{ij}, \tilde{\epsilon}^a_k v_{kj} + \tilde{\epsilon}^a_k \epsilon^v_{kj} + \epsilon^v_{kj}].
> $$
>
> Without independence, additional covariance terms will appear, but all contributions **remain weighted by $a_i^2$ or $a_ia_k$**. Consequently, tokens with large attention scores still dominate the variance of $\delta_j$, regardless of whether the covariance terms are zero or not.
>
> Therefore, removing the independence assumption does not change the claim that reconstruction error is dominated by tokens with large attention scores. Hence, even if the assumption in Eq. (8) is relaxed, the main insight and effectiveness of the proposed method remain valid.

---

### Official Review · Reviewer_dUD4 · 2025-10-31

**Soundness:** 3
**Presentation:** 2
**Contribution:** 2
**Rating:** 6
**Confidence:** 2

**Summary:**

This paper addresses the challenge of efficient deployment of Visual Autoregressive Models (VAR) by focusing on Post-Training Quantization (PTQ), a technique that enables deep network compression using a small subset of calibration data. While PTQ has shown promise in conventional diffusion models generative models, its application to VAR remains underexplored, primarily due to two critical issues:

* First, significant reconstruction errors arise from the multiplication of attention scores and value tokens in the VAR transformer, especially at coarse scales (low resolutions) where high attention scores are more concentrated—these errors propagate across subsequent finer scales and degrade final image quality.
* Second, limited calibration data leads to a mismatch between the sampling frequencies of VQVAE codebook entries and their predicted probabilities, biasing quantization parameters and reducing quantization performance.

To tackle these challenges, the paper proposes a PTQ framework tailored for VARs, consisting of two core components: Shift-and-Sum Quantization and Calibration Data Resampling.

Extensive experiments validate the framework on ImageNet across four tasks: class-conditional image generation, image in-painting, out-painting, and class-conditional editing. Evaluations on VARs of varying depths (16, 20, 24, 30 layers) and different bit-widths show consistent improvements over baseline methods (e.g., BRECQ, LiteVAR) in metrics like IS, gFID, and FID2FP16.

**Strengths:**

In fact, I have a good understanding of autoregressive models, but I am not an expert in the field of quantization. Please correct me if there are any issues with my descriptions.

1/ This paper mainly focuses on optimizing Post-Training Quantization for Visual Autoregressive Models. There is relatively little related work on autoregressive models, so this research is undoubtedly worthy of encouragement .

2/ This work has achieved promising results on ImageNet-256. It outperforms LiteVAR, and the performance improvement is even more significant when combined with LiteVAR.

3/ The analysis of "Reconstruction error across scales" is quite interesting. The authors found that quantization errors are more significant at early (coarse) scales, and based on this observation, they designed the Shift-and-Sum Quantization technique.

**Weaknesses:**

1/ I have a major question: Since the main purpose of this work is to improve the efficiency of generative models for deployment, why are there no experiments in the paper showing the speed performance or throughput performance of the VAR model after quantization?

2/ Currently, the experiments on VAR are only conducted at a resolution of 256. I am curious whether the results are consistent at higher resolutions. For example, at a resolution of 1024—admittedly, VAR itself has no experiments at 1024 resolution, but Infinity (the text-to-image model of VAR) has experiments at the 1024 resolution version, and it would be valuable to observe the experimental phenomena there.

reference:
Infinity: Scaling Bitwise AutoRegressive Modeling for High-Resolution Image Synthesis

**Questions:**

None

---

> ### Author Response · Authors · 2025-11-24
> **Response to Reviewer dUD4**
>
> We sincerely appreciate the constructive comments provided by Reviewer dUD4. Our detailed responses to the comments are presented below.
> *****
> > **W1. Throughput**
>
> We thank the reviewer for raising this important point. We agree that reporting throughput (images/sec) is highly relevant, given that the primary goal of our work is to improve the efficiency of VAR.
>
> A meaningful throughput measurement requires running the model with actual lower-precision arithmetic. Many PTQ papers, including the baselines in our comparison, rely on “fake quantization,” where values are rounded to quantized levels but the computation is still carried out in floating-point format. This is useful for evaluating accuracy, but it does not change the real computational cost.
>
> To measure real throughput, the model should be executed using genuine integer operations. At the moment, the official PyTorch quantization API does not support “true quantization” below 8 bits. For this reason, we are preparing an 8-bit quantized version of VAR and running it through ONNX Runtime, which allows the model to execute with real 8-bit computations. This will enable us to report actual images-per-second numbers rather than simulated behavior. We expect to include these results before the discussion deadline (Dec. 3).
> *****
> > **W2. Evaluation on Infinity, for text-to-image generation (1024x1024 resolution).**
>
> We thank the reviewer for the suggestion. We present the quantitative results for Infinity-2B [C1] in **Table R1**, which is also included in Table 2 of the revised paper. Following the experimental protocols in [C1], we exploit ImageReward [C2], HPSv2.1 [C3], and GenEval [C4], for evaluating our method. We can see that our method outperforms BRECQ and LiteVAR in various bit-widths, and applying our method on top of LiteVAR further boosts performance. In addition, we include the qualitative results for Infinity in Fig. C of the revised paper.
>
> [C1] Infinity: Scaling Bitwise AutoRegressive Modeling for High-Resolution Image Synthesis, CVPR 2025\
> [C2] Imagereward: Learning and evaluating human preferences for text-to-image generation, NeurIPS, 2023\
> [C3] Human preference score v2: A solid benchmark for evaluating human preferences of text-to-image synthesis, arXiv, 2023\
> [C4] Geneval: An object-focused framework for evaluating text-to-image alignment, NeurIPS, 2023
>
>
> **Table R1**: Quantitative comparison of quantizing Infinity-2B with various methods for text-to-image generation.
>
> | Bits (W/A) | Method          | ImageReward ↑ | HPSv2.1 ↑ | GenEval ↑ |
> |------------|------------------|----------------|------------|------------|
> | 16/16      | Full-precision   | 0.932          | 32.23      | 0.736      |
> | 6/6        | BRECQ            | 0.908          | 32.01      | 0.719      |
> | 6/6        | LiteVAR          | 0.912          | 32.01      | 0.720      |
> | 6/6        | Ours             | 0.914          | 32.03      | 0.722      |
> | 6/6        | **Ours+LiteVAR** | **0.929**      | **32.04**  | **0.725**  |
> | 4/8        | BRECQ            | 0.845          | 31.79      | 0.712      |
> | 4/8        | LiteVAR          | 0.866          | 32.03      | 0.719      |
> | 4/8        | Ours             | 0.861          | 31.94      | 0.716      |
> | 4/8        | **Ours+LiteVAR** | **0.880**      | **32.08**  | **0.722**  |
> | 4/6        | BRECQ            | 0.831          | 31.62      | 0.703      |
> | 4/6        | LiteVAR          | 0.838          | 31.61      | 0.702      |
> | 4/6        | Ours             | 0.842          | 31.70      | 0.706      |
> | 4/6        | **Ours+LiteVAR** | **0.880**      | **32.04**  | **0.714**  |
> | 4/4        | BRECQ            | 0.346          | 27.92      | 0.563      |
> | 4/4        | LiteVAR          | 0.407          | 28.98      | 0.626      |
> | 4/4        | Ours             | 0.575          | 29.32      | 0.653      |
> | 4/4        | **Ours+LiteVAR** | **0.748**      | **29.57**  | **0.672**  |

---

> ### Author Response · Authors · 2025-11-28
> **Response to Reviewer dUD4 - W1. Throughput**
>
> We export VAR-d16 to ONNX and apply 8-bit quantization to both weights and activations using ONNX Runtime, which executes genuine integer operations during inference. This enables us to report realistic images-per-second throughput rather than values obtained from fake quantization in floating-point arithmetic.
>
> We compare in **Table R2** (included in Table C of the revised paper) the throughput of VAR-d16 under different quantization settings. Quantizing the model to 8 bits substantially improves throughput (1.42–1.82×), confirming that reducing the actual numerical precision leads to meaningful gains in real execution speed. We also observe that our method is only about 2% slower than BRECQ, while being considerably faster than LiteVAR, highlighting the practical efficiency of our approach for real deployment scenarios.
>
> **Table R2**: Throughputs (image/sec) of VAR-d16 under different quantization methods, measured using ONNX
> Runtime on an Intel Xeon Gold 6226R CPU.
>
> | Method        | Bits (W/A) | Throughput (image/sec) |
> |---------------|------------|---------------------------|
> | Full-precision | 16/16      | 0.1208 ± 0.0010          |
> | BRECQ | 8/8        | 0.2195 ± 0.0046          |
> | LiteVAR | 8/8      | 0.1749 ± 0.0028          |
> | Ours          | 8/8        | 0.2147 ± 0.0042          |
> | Ours + LiteVAR | 8/8       | 0.1719 ± 0.0026          |

---

### Official Review · Reviewer_8oiH · 2025-11-10

**Soundness:** 3
**Presentation:** 3
**Contribution:** 2
**Rating:** 4
**Confidence:** 4

**Summary:**

This paper identifies two major challenges when applying post-training quantization (PTQ) to Visual Autoregressive Models (VAR): (1) large reconstruction errors arising from quantizing the multiplication between attention scores and value tokens, especially at coarse scales where high attention scores are more common; and (2) a mismatch between codebook-entry probabilities and their sampling frequencies during calibration due to limited calibration data. To address these issues, the authors propose two components: a shift-and-sum quantization technique that duplicates and symmetrically shifts attentive tokens (those with high attention scores) to reduce quantization errors, and a calibration data resampling method that reassigns codebook entries to better match predicted probabilities. Experiments across multiple VAR depths and tasks—including class-conditional generation, inpainting, outpainting, and conditional editing—show that the proposed methods consistently outperform prior PTQ methods while maintaining low computational overhead. The approach achieves state-of-the-art PTQ performance on VAR and is complementary to existing methods like LiteVAR.

**Strengths:**

- The two components proposed in the paper are well-designed to address the specific problems that arise in VAR quantization, and the paper clearly explains how they solve these issues.
- The proposed method appears broadly applicable beyond VAR, with potential usefulness in visual generation and autoregressive modeling in general.

**Weaknesses:**

- Given the nature of quantization research, more generic and widely applicable methods are preferable. However, the proposed approach is validated only on VAR, making the research scope narrow and potentially limiting its impact.
- Recent PTQ research on transformer quantization is not discussed; the related work mainly covers older studies. Similarly, LiteVAR also focuses specifically on VAR quantization, which suggests that the overall scope of related work is limited.

**Questions:**

- Could the proposed method be evaluated on other transformer-based models to verify whether it generalizes and improves performance? Although applying it to plain autoregressive generation may be less straightforward, architectures that use multi-scale representations might benefit significantly.
  - The following models might be worth exploring:
    - OneFormer: One Transformer to Rule Universal Image Segmentation, CVPR 2023
    - VGGT: Visual Geometry Grounded Transformer, CVPR 2025
- In BRECQ, the main PTQ evaluation table compares W4A4 and W2A4 settings. It would be interesting to see how the proposed method performs under these quantization settings compared to BRECQ. Can the authors provide results or insights on W4A4 and W2A4 performance?
- How does the proposed method perform quantitatively on inpainting, outpainting, and class-conditional editing tasks? Since the current version mainly focuses on quantization for VAR, it would be valuable to include numerical performance metrics for these tasks, rather than relying solely on qualitative visual comparisons.

---

> ### Author Response · Authors · 2025-11-24
> **Response to Reviewer 8oiH [1/3]**
>
> We sincerely appreciate the constructive comments provided by Reviewer 8oiH. Our detailed responses to the comments are presented below.
> *****
> > **W1, Q1. Evaluation on transformer-based models other than VAR**
>
> We appreciate the reviewer's suggestion and fully agree that evaluating our method on other transformer-based models—particularly those employing multi-scale representations such as OneFormer and VGGT—would provide valuable insights into the generality of our approach.
>
> We are preparing to quantize these models in the remaining discussion period and expect that quantitative results can be obtained before the final deadline (Dec. 3).
> *****
> > **W2. Related works**
>
> Thank you for pointing this out. We have revised Sec. 2 to include recent PTQ methods for transformers, such as PTQ4ViT [C1], RepQ-ViT [C2], IGQ-ViT [C3], and ERQ [C4], as follows:
> ```
> To date, a number of PTQ techniques have been developed for transformer architectures. For example, PTQ4ViT (Yuan et al., 2022) proposes twin uniform quantizers to address the long-tailed distributions of softmax attentions and activations after GELU (Hendrycks & Gimpel, 2016) nonlinearity. RepQ-ViT (Li et al., 2023b) proposes to consider inter-channel scale variations of activations after LayerNorm, and exploits a scale reparameterization technique that adjusts the affine factors of LayerNorm and the weights of subsequent fully-connected (FC) layers. IGQ-ViT (Moon et al., 2024) observes that input activations of FC layers vary drastically across input instances, and splits the channels of activation maps dynamically into multiple groups, quantizing the activation values within each group with a shared quantizer. ERQ (Zhong et al., 2025) proposes to adjust the rounding function for weights to minimize the quantization errors induced from activation quantization.
> ```
> These works primarily target ViT variants, whereas LiteVAR remains the only prior PTQ method specifically designed for VAR.
>
> [C1] Ptq4vit: Post-training quantization for vision transformers with twin uniform quantization, ECCV, 2022\
> [C2] Repq-vit: Scale reparameterization for post-training quantization of vision transformers, ICCV, 2023\
> [C3] Instance-aware group quantization for vision transformers, CVPR, 2024\
> [C4] Towards accurate post-training quantization of vision transformers via error reduction, TPAMI, 2025

---

> > ### Author Response · Authors · 2025-12-01
> > **Response to Reviewer 8oiH - W1, Q1. Evaluation on transformer-based models other than VAR**
> >
> > We appreciate the reviewer's insightful suggestion regarding the evaluation of our method on additional transformer-based architectures. To investigate the generalization capability of our approach beyond VAR, we have conducted experiments on OneFormer, a large-scale universal image segmentation model that relies heavily on multi-scale representations.
> >
> > As shown in **Table R4**, which is also included in Table D of the revised paper, our method consistently outperforms BRECQ and achieves performance comparable to LiteVAR, despite LiteVAR keeping the FC layers after the GELU non-linearity in full-precision. Moreover, combining our method with LiteVAR further improves quantized performance across semantic, instance, and panoptic segmentation tasks. These results provide strong evidence that our method generalizes well to transformer-based architectures beyond VAR.
> >
> >
> > **Table R4**: Quantitative comparison of quantizing OneFormer with various quantization methods for the tasks of semantic, instance, and panoptic segmentation on COCO.
> >
> > | **Bits (W/A)** | **Method**                  | **mIoU** | **AP** | **PQ** |
> > |----------------|------------------------------|---------:|-------:|-------:|
> > | 16/16    | Full-precision               | 67.4     | 49.0   | 57.9   |
> > | 6/6        | BRECQ (Li et al., 2021)      | 66.5     | 47.8   | 55.8   |
> > | 6/6       | LiteVAR (Xie et al., 2024)   | 66.9     | 48.2   | 57.4   |
> > | 6/6        | Ours                         | 66.8     | 48.3   | 57.6   |
> > | 6/6        | **Ours+LiteVAR**             | **67.0** | **48.6** | **57.8** |
> > | 4/4        | BRECQ (Li et al., 2021)       | 63.7     | 37.9   | 51.0   |
> > | 4/4        | LiteVAR (Xie et al., 2024)    | 64.4     | 39.2   | 51.5   |
> > | 4/4        | Ours                          | 64.6     | 39.1   | 51.6   |
> > | 4/4        | **Ours+LiteVAR**              | **65.0** | **40.6** | **52.2** |

---

> ### Author Response · Authors · 2025-11-24
> **Response to Reviewer 8oiH [2/3]**
>
> > **Q2. PTQ performances on W4A4 and W2A4 settings**
>
> We thank the reviewer for the suggestion. We have added the results for the W4A4 and W2A4 settings in **Table R1** (For W4A4 setting, we have also incorporated the result into Table 1 of the revised paper).
>
> - For W4A4, our method substantially outperforms both BRECQ and LiteVAR across all VAR architectures, and the performance gap becomes even larger when combined with LiteVAR. This demonstrates that the proposed shift-and-sum quantization is particularly effective at lower activation bit-widths.
>
> - For W2A4, we observe that all methods—including ours—exhibit severe degradation (**Table R2**). While we cannot rule out the possibility that future techniques may overcome this limitation, we are not aware of any prior PTQ work that reports successful 2-bit weight quantization for autoregressive or diffusion-based generative models while maintaining acceptable generation quality. This suggests that pushing VAR below 4-bit weights remains an open challenge.
>
> **Table R1**: Quantitative comparison of quantizing VAR-d16, 20, 24, and d30 with various methods for the task of conditional image generation under 4/4-bit setting.
>
> | Method        | IS(d16) | FID(d16) | FID2FP16(d16) | IS(d20) | FID(d20) | FID2FP16(d20) | IS(d24) | FID(d24) | FID2FP16(d24) | IS(d30) | FID(d30) | FID2FP16(d30) |
> |---------------|---------|----------|---------|---------|----------|---------|---------|----------|---------|---------|----------|---------|
> | BRECQ         | 67.6    | 33.03    | 32.82   | 94.1    | 23.03    | 22.42   | 119.0   | 16.89    | 17.07   | 125.3   | 15.71    | 14.66   |
> | LiteVAR       | 66.9    | 35.87    | 36.67   | 100.7   | 21.71    | 21.10   | 128.6   | 15.75    | 16.81   | 149.3   | 12.25    | 11.11   |
> | Ours          | 90.7    | 24.57    | 24.35   | 116.8   | 17.74    | 17.28   | 148.0   | 12.04    | 11.71   | 177.5   | 9.26     | 8.68    |
> | **Ours+LiteVAR** | **110.9** | **18.92** | **18.32** | **145.4** | **12.43** | **12.03** | **172.7** | **8.85** | **8.24** | **180.6** | **9.12** | **8.48** |
>
> **Table R2**: Quantitative comparison of quantizing VAR-d16, 20, 24, and d30 with various methods for the task of conditional image generation under 2/4-bit setting.
>
> | Method        | IS(d16) | FID(d16) | FID2FP16(d16) | IS(d20) | FID(d20) | FID2FP16(d20) | IS(d24) | FID(d24) | FID2FP16(d24) | IS(d30) | FID(d30) | FID2FP16(d30) |
> |---------------|---------|----------|----------------|---------|----------|----------------|---------|----------|----------------|---------|----------|----------------|
> | BRECQ         | 4.7     | 187.93   | 191.28         | 4.9     | 171.74   | 177.27         | 4.2     | 203.08   | 202.75         | 5.7     | 178.35   | 177.40         |
> | LiteVAR       | 4.1     | 192.14   | 197.54         | 5.6     | 172.15   | 177.25         | 6.0     | 170.94   | 175.32         | 4.8     | 181.33   | 186.35         |
> | Ours          | 5.4     | 174.50   | 177.07         | 5.8     | 170.06   | 174.27         | 5.9     | 145.50   | 149.42         | 5.9     | 173.32   | 175.88         |
> | **Ours+LiteVAR** | 5.7 | 178.86 | 181.65 | 5.7 | 169.51 | 173.67 | 6.1 | 122.16 | 120.27 | 5.5 | 153.22 | 156.45 |

---

> ### Author Response · Authors · 2025-11-24
> **Response to Reviewer 8oiH [3/3]**
>
> > **Q3. Quantitative results on the tasks of inpainting, outpainting, and class-conditional editing**
>
> We thank the reviewer for the helpful suggestion. In **Table R3**, we now provide quantitative results for image inpainting, outpainting, and class-conditional editing, which are also included in Table 3 of the revised paper.
>
> For image inpainting and outpainting, we follow standard evaluation protocols in prior generative inpainting literature (e.g., [C5], [C6]) and report LPIPS between the generated images and the corresponding ground-truth images.
>
> For class-conditional image editing, we compute CLIP scores between the edited images and the textual prompt "a photo of [class name]", where we denote [class name] as the target ImageNet category, following [C7] and [C8].
>
> These settings align with commonly used evaluation procedures in prior works, and the resulting metrics provide a clear quantitative comparison across different quantization methods. We hope that this addition sufficiently addresses the reviewer's request.
>
> [C5] Resolution-Robust Large Mask Inpainting With Fourier Convolutions, WACV 2022\
> [C6] A Task Is Worth One Word: Learning with Task Prompts for High-Quality Versatile Image Inpainting, ECCV 2024\
> [C7] Learning transferable visual models from natural language supervision, ICML 2021\
> [C8] LEDITS++: Limitless Image Editing using Text-to-Image Models, CVPR 2024
>
> **Table R3**: Quantitative comparison of quantizing VAR-d30 with various methods for the tasks of image inpainting, outpainting, and class-conditional editing.
>
> | Bits (W/A) | Method        | LPIPS ↓ (inpaint) | LPIPS ↓ (outpaint) | CLIP score ↑ (editing) |
> |------------|---------------|-------------------|--------------------|-------------------------|
> | 16/16      | Full-precision | 0.2827            | 0.2119             | 0.2480                  |
> | 6/6        | BRECQ          | 0.2852            | 0.2144             | 0.2455                  |
> | 6/6        | LiteVAR        | 0.2861            | 0.2143             | 0.2458                  |
> | 6/6        | Ours           | 0.2843            | 0.2141             | 0.2459                  |
> | 6/6        | **Ours+LiteVAR** | **0.2835**       | **0.2134**         | **0.2465**              |
> | 4/4        | BRECQ          | 0.2912            | 0.2207             | 0.2289                  |
> | 4/4        | LiteVAR        | 0.2907            | 0.2208             | 0.2318                  |
> | 4/4        | Ours           | 0.2888            | 0.2181             | 0.2358                  |
> | 4/4        | **Ours+LiteVAR** | **0.2879**       | **0.2172**         | **0.2363**              |

---

### Author Response · Authors · 2025-11-24
**Global response**

Dear AC and reviewers,

We sincerely appreciate your valuable time in reviewing our paper and providing constructive and insightful feedbacks.

In this work, we propose Shift-and-Sum Quantization, a novel PTQ framework tailored for visual autoregressive models (VAR). As highlighted by the reviewers, our method is well-motivated (8oiH, dUD4), interesting and technically sound (dUD4), extends naturally beyond VAR architectures (8oiH), and is supported by both theoretical analyses (dUD4, TesS, ND1G) and comprehensive empirical evaluations (dUD4, TesS). We are grateful for the recognition of the strengths of our contributions and for the helpful suggestions that guided our revisions.

We have carefully addressed all reviewer comments and substantially improved the paper. All modifications in the revised paper are highlighted in red for ease of verification. Below, we summarize the major changes:

- Added recent PTQ methods for transformers (Sec. 2)
- PTQ performances for the 4/4-bit setting (Table 1)
- Results for quantizing Infinity (Table 2)
- Quantitative results for inpainting, outpainting, and class-conditional editing (Table 3)
- Discussion on the independence assumption for quantization noise (Table A)
- Analysis of computational overheads (Table B)
- Qualitative comparison across bit-widths (Fig. E)
- IS–BOP and FID–BOP trade-off curves (Fig. F)
- Measurement of throughput after quantization (Table C)
- Evaluation on transformer-based models beyond VAR (Table D)

We hope the revisions meaningfully strengthen the paper and address all reviewers' concerns.
We sincerely appreciate the thoughtful feedback and look forward to further discussion.

Sincerely,\
Authors

---

### Meta-Review · Area_Chair_kGJP · 2026-01-15

**Summary:**

This paper studies post-training quantization for visual autoregressive models, and identifies two VAR-specific failure modes: (i) amplified reconstruction error in attention–value products at coarse scales, and (ii) mismatch between VQ codebook sampling frequencies and predicted probabilities. The authors propose a shift-and-sum quantization mechanism and a calibration resampling strategy to solve these limitations.  Empirically, the method improves PTQ quality across VAR depths and tasks, and the rebuttal adds evidence on higher-resolution Infinity and non-VAR transformer (OneFormer) settings.

The initial ratings of this paper include three weak accept and one weak reject. The major concerns of reviewer 8oiH (who gave 4) were directly addressed in the rebuttal. This paper presents both (i) a concrete technical mechanism (shift-and-sum) targeting a documented failure mode (coarse-scale attention–value amplification) and (ii) a simple but impactful calibration fix (probability-aligned resampling), validated across multiple VAR depths and tasks. Given the technical strengths and the reviewers' recommendation, the AC agree to accept this paper.

**Reviewer Concerns:**

We mainly discuss the review for reviewer 8oiH as they are the only one that recommend rejection. Reviewer 8oiH (rating 4: below threshold; asked for broader scope + missing related work + extra quant/task results).

Addressed by rebuttal:
- Broader scope / generalization beyond VAR: They added an evaluation on OneFormer (multi-scale transformer) and show comparable or improved performance vs baselines and gains when combined with LiteVAR.
- Missing recent transformer PTQ related work: They explicitly add recent PTQ-for-transformer citations (PTQ4ViT / RepQ-ViT / IGQ-ViT / ERQ) in the response.
- W4A4 / W2A4 comparison: They provide W4A4 and W2A4 results and explain that W2A4 remains severely degraded for all methods.
- Quantitative metrics for in/out-painting and editing: They add LPIPS for (in/out)-painting and CLIP score for editing with a full table.

Still outstanding / partially addressed:
- Breadth remains limited: only one non-VAR architecture (OneFormer) is shown; suggested models like VGGT are not evaluated (or at least not evidenced in the provided discussion).
- W2A4 practicality: they acknowledge the limitation but it remains a “known-hard” regime rather than a resolved point.

**Reviewer Scores:**

- Reviewer 8oiH (4 → likely 6): The reviewer’s main blockers (scope narrow, missing related PTQ work, missing W4A4/W2A4 and quantitative in/out/edit metrics) are directly addressed with new experiments/tables and related-work additions

- Reviewer dUD4 (6 → likely 7): Their two primary weaknesses (no throughput; no higher-res test) are resolved with an ONNX 8-bit throughput table and Infinity-2B results.

- Reviewer TesS (6 → likely 6–7): If their main hesitation were the Eq. 8 independence assumption, the authors’ clarification + empirical near-zero correlation evidence would likely raise confidence in soundness

- Reviewer ND1G (6 → likely 6): Their three concrete asks (overhead analysis, tradeoff/bitwidth visualization, and better metrics) are answered.

---

### Decision · Program_Chairs · 2026-01-26

Accept (Poster)